# UNDERSTANDING AND TACKLING OVER-DILUTION IN GRAPH NEURAL NETWORKS

## ABSTRACT

Message Passing Neural Networks (MPNNs) have become the predominant architecture for representation learning on graphs. While they hold promise, several inherent limitations have been identified, such as over-smoothing and over-squashing. Both theoretical frameworks and empirical investigations substantiate these limitations, facilitating advancements for informative representation. In this paper, we investigate the limitations of MPNNs from a novel perspective. We observe that even in a single layer, a node's own information can become considerably diluted, potentially leading to negative effects on performance. To delve into this phenomenon in-depth, we introduce the concept of *Over-dilution* and formulate it with two types of dilution factors: *intra-node dilution* and *inter-node dilution*. *Intra-node dilution* refers to the phenomenon where attributes lose their influence within each node, due to being combined with equal weight regardless of their practical importance. *Inter-node dilution* occurs when the node representations of neighbors are aggregated, leading to a diminished influence of the node itself on the final representation. We also introduce a transformer-based solution, which alleviates over-dilution by merging attribute representations based on attention scores between node-level and attribute-level representations. Our findings provide new insights and contribute to the development of informative representations.

## 1 INTRODUCTION

Recent progress in representation learning on graph-structured data has been largely attributed to Graph Neural Networks (GNNs), powered by their ability to utilize structural information. In particular, Message Passing Neural Networks (MPNNs) have gained significant attention due to their simple mechanism yet powerful performance (Gilmer et al., 2017). Various extensions of MPNNs have been proposed, primarily, to improve their expressivity and solve issues with degeneration caused during the message passing (Kipf & Welling, 2017; Veličković et al., 2018; Hamilton et al., 2017; Wu et al., 2019; Chen et al., 2020b; Corso et al., 2020; Bianchi et al., 2021; Brody et al., 2022).

Towards a deeper understanding, several phenomena have been observed and formalized that cause MPNNs to deviate from optimal behavior, such as over-smoothing (Xu et al., 2018; Li et al., 2018b; Nt & Maehara, 2019; Zhao & Akoglu, 2020; Oono & Suzuki, 2020; Chen et al., 2020a), over-squashing (Alon & Yahav, 2021; Topping et al., 2022), and over-correlation (Jin et al., 2022). They have become the foundation for addressing distortions in information on irregular structures, laying the groundwork for subsequent studies to enhance MPNNs (Arnaiz-Rodríguez et al., 2022; Wu et al., 2023; Guo et al., 2023; Eliasof et al., 2023; Nguyen et al., 2023; Di Giovanni et al., 2023; Karhadkar et al., 2023; Gravina et al., 2023). Therefore, it is essential to identify and formalize the limitations (i.e. undesirable behaviors) of MPNNs for the advancement of representation learning on graphs.

In this paper, we investigate a limitation associated with the *preservation* of *attribute-level* information. This perspective is distinct from previous categories of limitations, where the primary focus has been on the *propagation* of *node-level* representation as illustrated in Figure 1. Although often not emphasized sufficiently, node attributes provide important information about the nodes that can be used to make predictions such as potential links between them (Gong et al., 2014; Huang et al., 2017; Li et al., 2017; 2018a; Hao et al., 2021). We first introduce the phenomenon that outlines the diminishment of a node's own information on the final representation in MPNNs, referred to as *over-dilution*. This phenomenon has been observed when nodes have an excessive number of

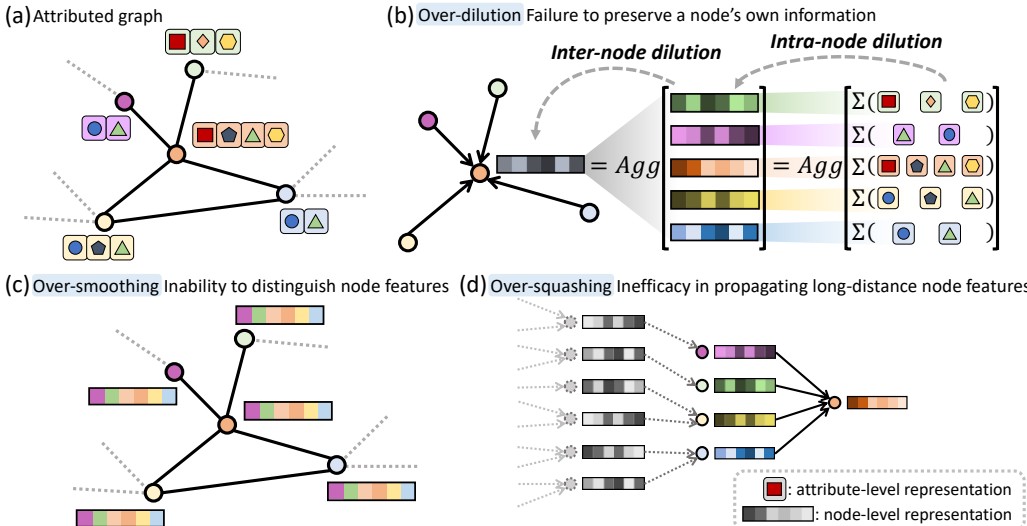

Figure 1: (a) illustrates an attributed graph. Each node is initially characterized by a set of attributes, which are represented by corresponding vectors. (b) illustrates the two sequential steps of the dilution process occurring in the first layer of MPNNs. The first step, intra-node dilution, occurs when the node-level representation is formed by summing the representations of its attributes. The more attributes a node has, the less influence each attribute exerts on the node's representation. The second step, inter-node dilution, takes place when a node representation is integrated with those of neighbor nodes during message propagation. (c) and (d) illustrate over-smoothing and over-squashing.

attributes, hindering their ability to focus on important attributes, or when each node receives an overwhelming amount of information from neighboring nodes, leading to a relative loss of their individual information. As illustrated in Figure 1, we analyze this phenomenon by dividing it into two cascaded sub-phenomena: *intra-node dilution* and *inter-node dilution*. These describe the weakening of influence of the attribute-level and the node-level representations, respectively.

To address the over-dilution phenomenon, we introduce a transformer-based architecture (Vaswani et al., 2017) designed to utilize attribute representations as tokens. Notably, this architecture is not a competitor but a complement to existing node embedding methods (e.g. MPNNs). Its flexibility is underscored by its ability to seamlessly integrate with any node embedding method, computing the final representation by weighting attribute representations based on attention scores associated with the aggregated node-level representation. We theoretically and empirically demonstrate its effectiveness for solving the over-dilution problem. Our main contributions can be summarized as:

- We introduce the *over-dilution* phenomenon from a new perspective, shedding light on its impact on the representation of graph-structured data. We formulate and elucidate this concept through two sub-phenomena: *intra-node dilution* and *inter-node dilution*, which describe the dilution of attribute-level and node-level representations, respectively.

- The concept of over-dilution delves into the limitation tied to the *preservation* of *attribute-level* information, setting it apart from existing limitations primarily centered on the *propagation* of *node-level* representation.

- By investigating the over-dilution phenomenon and addressing it with a transformer-based approach that complements any node embedding methods, we contribute to a deeper understanding and provide insights into the development of informative representations.

## 2 PRELIMINARIES

Attributed graphs are of the form $\mathcal{G} = (\mathcal{T}, \mathcal{V}, \mathcal{E})$ that consists of sets of attributes $\mathcal{T}$, nodes $\mathcal{V}$, and edges $\mathcal{E} \subseteq \mathcal{V} \times \mathcal{V}$. Let $\mathcal{T}_v$ be a subset for attributes $t \in \mathcal{T}$ that node $v \in \mathcal{V}$ is associated with. $N_{\mathcal{V}} = |\mathcal{V}|$ and $N_{\mathcal{T}} = |\mathcal{T}|$ indicate the total numbers of nodes and attributes, respectively. The

node feature matrix and the adjacency matrix are represented as $X \in \mathbb{R}^{N_\mathcal{V} \times N_\mathcal{T}}$ and $A \in \mathbb{R}^{N_\mathcal{V} \times N_\mathcal{V}}$, respectively. We assume that each node has a discrete binary vector, $X_v \in \mathbb{R}^{N_\mathcal{T}}$, indicating existence of attributes where $X_{v,t}$=1 if the node $v$ has attribute $t$, otherwise $X_{v,t}$=0. The embedding of attribute $t$ is represented as $z_t \in \mathbb{R}^d$ with dimension $d$, which is a randomly initialized representation.

## 2.1 MESSAGE PASSING NEURAL NETWORKS

In MPNNs, the representations of nodes are calculated through a series of layers, where each layer consists of two main operations: the *Update* function and the *Aggregate* function. The update function is used to transform the node representation and the aggregate function is used to combine information from neighboring nodes. This process is repeated for multiple layers, thereby refining the node representations and extracting higher-level features from the graph. We formulate MPNNs as:

$$h_v^{(l)} = \sigma(\text{Aggregate}(\{\text{Update}(h_u^{(l-1)})|u \in \tilde{\mathcal{N}}(v)\})) = \sigma(\sum_{u \in \tilde{\mathcal{N}}(v)} \alpha_{vu} h_u^{(l-1)} W^{(l)}) \tag{1}$$

where $\tilde{\mathcal{N}}(v)$ is a set of neighbor nodes of $v$ including itself, $W^{(l)} \in \mathbb{R}^{d \times d}$ is the learnable parameter at $l$-th layer, and Aggregate($\cdot$) denotes the aggregate function for neighbor nodes. $h^{(0)} = XW^{(0)} \in \mathbb{R}^{N_\mathcal{V} \times d}$ is the initial node feature matrix with learnable parameter $W^{(0)} \in \mathbb{R}^{N_\mathcal{T} \times d}$ and dimension $d$. In this context, the $t$-th row of $W^{(0)}$ is equivalent to $z_t$, the representation of the corresponding attribute. The parameter $\alpha_{vu}$ denotes the aggregation coefficient assigned to the edge connecting neighbor node $u$ to the center node $v$ in the aggregation function. This coefficient is calculated as $\frac{1}{\sqrt{\deg(v)\deg(u)}}$ in the case of GCN, or as an attention coefficient between nodes $v$ and $u$ in GAT. The receptive field of node $v$ is defined as: $\mathcal{B}_l(v) := \{u \in \mathcal{V} \mid s_\mathcal{G}(v,u) \leq l\}$, where $s_\mathcal{G}$ is the standard shortest-path distance on the graph $\mathcal{G}$ and $l \in \mathbb{N}$ is the radius.

## 2.2 OVER-SMOOTHING AND OVER-SQUASHING

Over-smoothing refers to the phenomenon where the model excessively propagates information between nodes, leading to a loss of distinguishability of their representations (Xu et al., 2018; Nt & Maehara, 2019; Oono & Suzuki, 2020). In the process of exchanging information through message propagation, all nodes have similar representations and noise is conveyed alongside important information (Li et al., 2018b; Chen et al., 2020a).

Over-squashing is a problem that arises when exponentially increasing amounts of information are compressed into a fixed-size vector (Alon & Yahav, 2021). This leads to a bottleneck, particularly in the extended paths within a graph, which hinders GNNs from fitting long-range signals and causes them to fail to propagate messages originating from distant nodes. As a result, the performance is typically compromised, where the task necessitates long-range interaction (Topping et al., 2022).

## 3 OVER-DILUTION

In this section, we introduce a new concept named over-dilution, which is distinct from over-smoothing and over-squashing as illustrated in Figure 1. Over-dilution refers to the diminishment of a node's information at both the attribute and the node levels. To assess the severity of over-dilution, we define the dilution factor, as a metric that measures the retention of node's own information in the updated representation. This factor can be decomposed into two cascaded components as described in Eq (2). We define the intra-node dilution factor $\delta_v^{\text{intra}}$ mainly for attribute representations and the inter-node dilution factor $\delta^{\text{inter}}$ for node representations. As depicted in Figure 1 (b), the attribute representation is diluted during the first step of message passing (i.e. *Update*) and then subsequently diluted in the second step (i.e. *Aggregate*) in form of the node-level representation. Therefore, the dilution factor of attribute $t$ at node $v$ can be defined as corresponding to two cascaded steps:

$$\delta_{v,t} = \delta_v^{\text{intra}}(t) * \delta^{\text{inter}}(v) \tag{2}$$

where $\delta_v^{\text{intra}}(t)$ represents the intra-node dilution factor of attribute $t$ at the node $v$ and $\delta^{\text{inter}}(v)$ represents the inter-node dilution factor of node $v$ in the graph. We exploit the *Jacobian matrix* of node representations to quantify dilution factors based on the influence distribution in a similar way as Xu et al. and Topping et al..

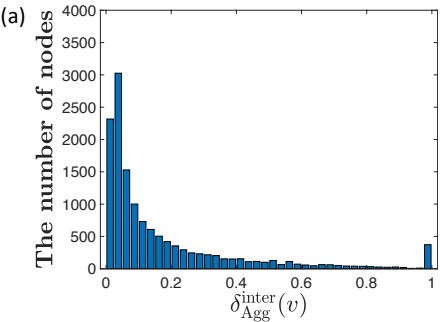 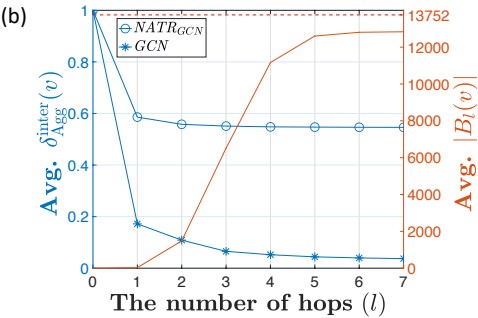

Figure 2: (a) The histogram of the inter-dilution factor (aggregation-only) values after single layer of *GCN* in Computers dataset. (b) The average of the inter-dilution factor (aggregation-only) with the left $y$-axis and the average size of the receptive field with the right $y$-axis in the Computers dataset.

Taking the Computers dataset as a primary example, consider a node with 204 attributes (the median value for the number of attributes) and 19 neighboring nodes (the median degree). In this scenario, each attribute of the node would be diluted to $1/204 * 1/20$, or roughly 0.025%, in a single layer when using either the *mean* or *sum* as the aggregation operator.

## 3.1 Intra-Node Dilution: Measuring Attribute Influence within Each Node

The intra-node dilution factor is a metric that quantifies the degree to which an attribute is diluted at a specific node. We measure the influence of $z_t$ on $h_v^{(0)}$ indicating how much the representation of attribute $t$ affects the initial representation of node $v$.

**Definition 3.1.** *(Intra-node dilution factor). For a graph $\mathcal{G} = (\mathcal{T}, \mathcal{V}, \mathcal{E})$, let $z_t$ be the representation of attribute $t \in \mathcal{T}$ and $h_v^{(0)}$ denote the initial feature representation of node $v \in \mathcal{V}$, which is calculated from the representations of attribute subset $\mathcal{T}_v$ that node $v$ possesses. The influence score $I_v(t)$ attribute $t$ on node $v$ is the sum of the absolute values of the elements in the Jacobian matrix $\left[\frac{\partial h_v^{(0)}}{\partial z_t}\right]$. We define the intra-node dilution factor as the influence distribution by normalizing the influence scores: $\delta_v^{intra}(t) = I_v(t)/\Sigma_{s \in \mathcal{T}_v} I_v(s)$. In detail, with the all-ones vector $e$:*

$$\delta_v^{intra}(t) = e^T \left[\frac{\partial h_v^{(0)}}{\partial z_t}\right] e \left/ \sum_{s \in \mathcal{T}_v} e^T \left[\frac{\partial h_v^{(0)}}{\partial z_s}\right] e \right. \tag{3}$$

**Hypothesis 1.** *(Occurrence of intra-node dilution). Intra-node dilution occurs when a node-level representation is computed by equally weighting and fusing attribute-level representations, irrespective of their individual importance. The over-dilution effect at the intra-node level becomes more pronounced as the number of attributes increases.*

For example, given node $v$ where the important attributes are sparse compared to the total number of attributes $|\mathcal{T}_v|$, the influence of the key attributes get proportionally limited to $1/|\mathcal{T}_v|$. In MPNNs, the representation of node $v$ is calculated by summing or averaging the representations of attributes $t \in \mathcal{T}_v$ as $h_v^{(0)} = XW^{(0)} = \sum_{t \in \mathcal{T}_v} z_t$ or $h_v^{(0)} = \sum_{t \in \mathcal{T}_v} \frac{z_t}{|\mathcal{T}_v|}$. Therefore, the intra-node dilution factor $\delta_v^{intra}(t)$ takes the constant value $\frac{1}{|\mathcal{T}_v|}$ for all attributes at each node $v$. This implies that the influence of each attribute on the representation of a node is treated as equal and, as the number of attributes increases, the impact of important attributes on the representation of the node is diluted. Given that attributes possess different levels of practical importance, their influences may be diluted in cases where only a small subset of attributes is crucial for the node representation.

## 3.2 Inter-node Dilution: Measuring Node Influence on Final Representation

The inter-node dilution factor of each node is calculated by considering the influence of the initial node representation on the output representation in the last layer and the influences of all other nodes. We adapt the Jacobian matrix of node representation, as introduced by Xu et al. for quantifying the influence of one node on another, to measure the influence of each node on itself.

**Definition 3.2.** *(Inter-node dilution factor). Let $h_v^{(0)}$ be the initial feature and $h_v^{(l)}$ be the learned representation of node $v \in \mathcal{V}$ at the $l$-th layer. We define the inter-node dilution factor as the normalized influence distribution of node-level representations: $\delta^{inter}(v) = I_v(v)/\Sigma_{u \in \mathcal{V}} \, I_v(u)$, or*

$$\delta^{inter}(v) = e^T \left[ \frac{\partial h_v^{(l)}}{\partial h_v^{(0)}} \right] e \bigg/ \sum_{u \in \mathcal{V}} e^T \left[ \frac{\partial h_v^{(l)}}{\partial h_u^{(0)}} \right] e \tag{4}$$

In MPNNs, the representation $h_v^{(l)}$ is calculated from the non-linear transformation (i.e. Update$(\cdot)$) and the aggregation of the representations $h_u^{(l-1)}$ for $u \in \tilde{N}(v)$. To observe the effect of the aggregation exclusively, we eliminate the effect of the non-linear transformation by setting all weight and initial node feature matrices to be the identity matrix. We define $\delta_{\text{Agg}}^{\text{inter}}(v)$, which is the exclusive version of the inter-node dilution factor, with $W^{(l)} = W^{(l-1)} = ... = W^{(1)} = h^{(0)} = I_{N_\mathcal{V}} \in \mathbb{R}^{N_\mathcal{V} \times N_\mathcal{V}}$. The output representation of the aggregation-only version of $GCN$ is calculated as $h'^{(l)} = (\tilde{D}^{-\frac{1}{2}} \tilde{A} \tilde{D}^{-\frac{1}{2}})^l I_{N_\mathcal{V}}$, where $\tilde{A}$ indicates the adjacency matrix with self-loop and $\tilde{D}$ is the corresponding degree matrix. The numerator of $\delta_{\text{Agg}}^{\text{inter}}(v)$ is calculated from:

$$\frac{\partial h_v'^{(l)}}{\partial h_v^{(0)}} = \underbrace{\prod_{i=1}^{l} \alpha_{vv}^{(i)} \cdot \frac{\partial h_v^{(0)}}{\partial h_v^{(0)}}}_{\text{for } l \geq 1} + \underbrace{\sum_{u \in \tilde{\mathcal{N}}(v) \backslash \{v\}} \sum_{k=1}^{l-1} \left( \prod_{\substack{j=k+2 \\ k \leq l-2}}^{l} \alpha_{vv}^{(j)} \right) \alpha_{vu}^{(k+1)} \frac{\partial h_u'^{(k)}}{\partial h_v^{(0)}}}_{\text{for } l \geq 2} \tag{5}$$

where $\alpha_{vu}^{(i)}$ indicates the aggregation coefficient from node $u$ to the node $v$ at $i$-th layer. The former term, which is defined for $l \geq 1$, indicates the preserved amount of the representation of node $v$ and the latter term, which is defined for $l \geq 2$, indicates the returned amount of representation of node $v$ from neighbors after more than two hops aggregation. The denominator of $\delta_{\text{Agg}}^{\text{inter}}(v)$ is calculated from:

$$\sum_{u \in \mathcal{V}} \frac{\partial h_v'^{(l)}}{\partial h_u^{(0)}} = \sum_{x \in \tilde{N}(v)} \sum_{u \in \mathcal{V}} \sum_{k=0}^{l-1} \left( \prod_{\substack{j=k+2 \\ k \leq l-2}}^{l} \alpha_{vv}^{(j)} \right) \alpha_{vx}^{(k+1)} \frac{\partial h_x'^{(k)}}{\partial h_u^{(0)}} \tag{6}$$

**Hypothesis 2.** *(Occurrence of inter-node dilution 1). For a node $v$ and its adjacent nodes, which are denoted as $\tilde{\mathcal{N}}(v)$, inter-node dilution occurs when the aggregation coefficient of the self-loop, $\alpha_{vv}$, is significantly smaller than the sum of the coefficients of the other edges connecting node $v$ and its adjacent nodes: $\alpha_{vv} \ll \sum_{u \in \tilde{\mathcal{N}}(v) \backslash \{v\}} \alpha_{vu}$.*

The inter-node dilution factor for the aggregation-only at a single layer is calculated as:

$$\delta_{\text{Agg}}^{\text{inter}}(v) = e^T \left[ \alpha_{vv} \frac{\partial h_v^{(0)}}{\partial h_v^{(0)}} \right] e \bigg/ e^T \left[ \sum_{u \in \tilde{N}(v)} \alpha_{vu} \frac{\partial h_u^{(0)}}{\partial h_u^{(0)}} \right] e = \frac{\alpha_{vv}}{\sum_{u \in \tilde{N}(v)} \alpha_{vu}} \tag{7}$$

In most MPNNs, the inter-node dilution occurs when the degree (i.e. $|\mathcal{N}(v)|$) is high. For $GCN$, it can even occur with the low degree if the neighbor nodes have smaller degrees compared to node $v$, because the aggregation coefficient for self-loop is defined as $\alpha_{vv} = \frac{1}{\deg(v)}$ while the coefficients for edges with neighbor nodes are defined as $\alpha_{vu} = \frac{1}{\sqrt{\deg(v)\deg(u)}}$. As shown in the Figure 2(a), a significant number of nodes exhibits low $\delta_{\text{Agg}}^{\text{inter}}(v)$ values even after one-hop aggregation.

**Hypothesis 3.** *(Occurrence of inter-node dilution 2). Inter-node dilution occurs at node $v$ as the size of its receptive field $|\mathcal{B}_l(v)|$ increases.*

As explained in Xu et al. and Topping et al., the size of the receptive field grows exponentially as the number of layers increases. Consequently, the information from a larger number of nodes is integrated, resulting in a dilution of the information specific to each individual node. Figure 2(b) illustrates the average of this relationship between the inter-node dilution factor (aggregation-only) and the average size of the receptive field in the Computers dataset, as the number of hops increases.

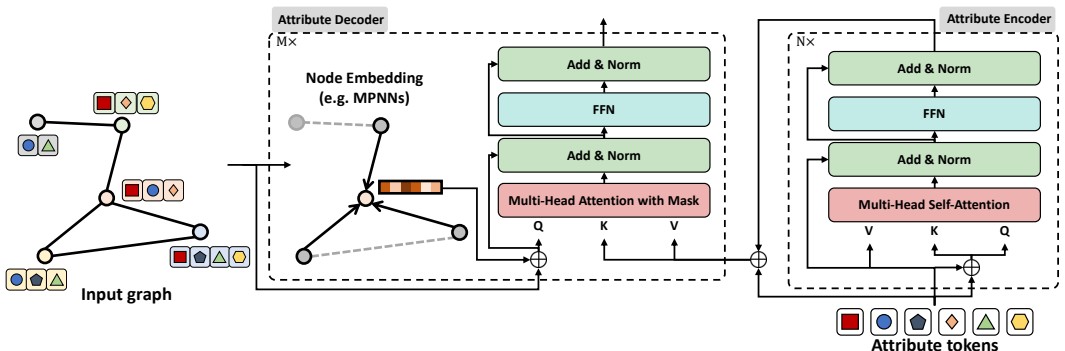

Figure 3: The overall architecture of the Node Attribute Transformer (*NATR*) comprises of two main components: the attribute encoder for attribute-level representations and the attribute decoder for node-level representations. It can be combined with any node embedding modules such as MPNNs.

## 4 NODE ATTRIBUTE TRANSFORMER

In this section, we describe the architecture of Node Attribute Transformer (*NATR*) in details. As illustrated in Figure 3, *NATR* consists of the attribute encoder and the attribute decoder. While the encoder is designed to consider the correlation between attributes, the decoder plays a crucial role in mitigating over-dilution. It integrates attribute representations across all layers, addressing inter-node dilution, and assigns greater weight to important attributes, tackling intra-node dilution, as discussed in Section 6.1.

### 4.1 ATTRIBUTE ENCODER

Given a set of randomly initialized representations of attribute tokens $z_t^{(0)} \in \mathbb{R}^{d_\mathcal{T}}$ and its matrix form $Z^{(0)} \in \mathbb{R}^{N_\mathcal{T} \times d_\mathcal{T}}$ with $d_\mathcal{T}$ dimension, the attribute representation $Z^{(n)}$, which is the output of $n$-layer of the attribute encoder, is obtained as: $Z^{(n)} = \text{SelfAttn}^{(n)}(Z^{(n-1)})$, where $\text{SelfAttn}^{(n)}$ is the $n$-th layer of the attribute encoder containing Multi-Head Self-Attention (MHSA), Add&Norm (Ba et al., 2016), and Feed-Forward Network (FFN) layers as illustrated in the Figure 3. We add $z_t^{(0)}$ to $z^{(n)}$ for the key and the query at all encoder layers like the positional encoding and it is omitted in the formulation for simplicity. After $N$ layers of attribute encoder in total, the attribute representation $z_t = z_t^{(N)} + z_t^{(0)}$, which is $Z \in \mathbb{R}^{N_\mathcal{T} \times d_\mathcal{T}}$ in a matrix form, is fed to the attribute decoder. For simplicity, we use the same dimension ($d_\mathcal{T} = d$) for attribute-level and node-level representations.

### 4.2 ATTRIBUTE DECODER

The attribute decoder is comprised of the node embedding module, Multi-Head Attention (MHA), Add&Norm, and FFN. The output of the attribute encoder, $Z$ is used to calculate the key $K^{(m)} = ZW_{DEC,K}^{(m)}$ and the value $V^{(m)} = ZW_{DEC,V}^{(m)}$ in the MHA of $m$-th decoder layer. The query $Q^{(m)} = H^{(m)}W_{DEC,Q}^{(m)}$ is calculated from the output of the node embedding module $H^{(m)} \in \mathbb{R}^{N_\mathcal{V} \times d}$, such as MPNNs, at the $m$-th decoder layer:

$$H^{(m)} = NodeModule(\tilde{H}^{(m-1)}, A) \tag{8}$$

where $\tilde{H}^{(m-1)}$ is the output of the previous decoder layer ($\tilde{H}^{(0)} = H^{(0)} = XW^{(0)}$) and $A$ represents the adjacency matrix. We add $H^{(0)}$ before calculating the query at all decoder layers and it is also omitted in the formulation for simplicity. We denote the node embedding module in the subscript as *NATR$_{NodeModule}$*. If $H^{(m)}$ is updated by $GCN$ layer with the formulation $H^{(m)} = \tilde{D}^{-\frac{1}{2}}\tilde{A}\tilde{D}^{-\frac{1}{2}}\tilde{H}^{(m-1)}W^{(m)}$, the model is denoted as *NATR$_{GCN}$*. Then, the attention coefficient for each attribute at MHA is calculated according to the node-level representation $Q^{(m)}$.

$$O^{(m)} = \text{MHA}(Q^{(m)}, K^{(m)}, V^{(m)}) = \text{Concat}(\text{head}_1, ..., \text{head}_h)W_{DEC,O}^{(m)} \tag{9}$$

Table 1: Dataset statistics. $|\mathcal{T}_v|$ and degree are related to intra- and inter-node dilutions, respectively.

| | $|\mathcal{V}|$ | Avg. Degree | Median Degree | $|\mathcal{T}|$ | Avg. $|\mathcal{T}_v|$ | Median $|\mathcal{T}_v|$ | Max. $|\mathcal{T}_v|$ |
|---|---|---|---|---|---|---|---|
| Amazon Computers | 13752 | 30.393 | 19 | 767 | 267.2 | 204 | 767 |
| Amazon Photo | 7650 | 26.462 | 18 | 745 | 258.8 | 193 | 745 |
| Cora ML | 2995 | 4.632 | 3 | 2879 | 50.5 | 49 | 176 |
| OGB-DDI$_{\text{SUBSET}}$ | 3531 | 499.582 | 500 | 1024 | 58.2 | 56 | 270 |
| OGB-DDI$_{\text{FULL}}$ | 4267 | 500.544 | 446 | 1024+1 | 49.1 | 51 | 271 |

where $\text{head}_i = \text{softmax}(\frac{Q_i K_i^\top}{\sqrt{d}})V_i$. We use masks in the MHA to merge the representations of the attributes possessed by each node exclusively. The aggregated representation of neighbor nodes $H^{(m)}$ is added to the output $O^{(m)} \in \mathbb{R}^{N_\mathcal{V} \times d}$ and then fed to Normalization layer followed by FFN layer. After additional normalization layer with skip connection, the final representation at $m$-th layer of attribute decoder, $\tilde{H}^{(m)} \in \mathbb{R}^{N_\mathcal{V} \times d}$ is used as the input feature of the node embedding module at the next decoder layer.

$$G^{(m)} = \text{Norm}(H^{(m)} + O^{(m)}), \quad \tilde{H}^{(m)} = \text{Norm}(\text{FFN}(G^{(m)}) + G^{(m)}) \tag{10}$$

Note that $H^{(m)}$ is the representation which is aggregated from neighbors and $O^{(m)}$ is the representation of each node. Therefore, we can control the inter-node dilution factor by changing as $G^{(m)} = \text{Norm}((1 - \lambda)H^{(m)} + \lambda O^{(m)})$, where $0 \leq \lambda \leq 1$ can be a hyperparameter, a learnable parameter, or an attention coefficient. In this work, we use the original formulation in Eq. (10).

**Plug-in Version of NATR.** We also provide a variant of *NATR* which is a plug-in version. In the scenario where an established node embedding model (e.g. MPNNs) is already trained and running in the industry, *NATR* can be easily incorporated into the existing model as a form of the separate architecture which is illustrated in Appendix. The plug-in version is also available for single layer models such as SGC (Wu et al., 2019). The difference from standard *NATR* is that the node embedding module is not nested inside the decoder but operated separately.

## 5 Experiments

We evaluate *NATR* on four benchmark datasets with the OGB pipeline (Hu et al., 2020). To validate the informativeness of node representations, we conduct the link prediction and the node classification tasks. All experiments are repeated 20 times, and the averages of performance are reported. *GCN* (Kipf & Welling, 2017), *SGC* (Wu et al., 2019), and *GAT* (Veličković et al., 2018) are selected as the main baselines. For *NATR$_{SGC}$*, we adapt the plug-in version of the architecture. In *GCN* and *SGC*, aggregation coefficients are calculated based on the degree. *GAT* uses attention coefficients to calculate its aggregation coefficients. Thereby, the inter-node dilution factor in *GCN* and *SGC* is affected by the topology, while in *GAT* it is affected by node representations. The details of the experiments, the extended results, a comparison with various node embedding modules such as GCNII and Graphormer (Hamilton et al., 2017; Bianchi et al., 2021; Corso et al., 2020; Chen et al., 2020b; Ying et al., 2021), an analysis on complexity, and ablation studies are reported in the Appendix.

### 5.1 Datasets

Computers and Photo datasets are segments of the Amazon co-purchase graph (McAuley et al., 2015; Shchur et al., 2018). Nodes indicate products and edges represent that two products are purchased together frequently. The bag-of-words in the product reviews is used as a set of attributes. CoraML dataset also contains bag-of-words as attributes, but in this case, nodes are documents and edges represents the citation link between them (McCallum et al., 2000; Bojchevski & Günnemann, 2018). In the OGB-DDI dataset, provided by Wishart et al. and Hu et al., each node represents drug and the edges represent interaction between drugs. We extract node attributes from molecular structures in DrugBank DB (Wishart et al., 2018) and generating Morgan Fingerprints (radius 3, 1024 bits) with RDKit. Any nodes that are not supported by RDKit or DrugBank are deleted, and the corresponding graph is subsequently reconstructed as OGB-DDI$_{\text{SUBSET}}$. The OGB-DDI$_{\text{FULL}}$ dataset includes all nodes and edges and the unsupported nodes are assigned a dummy attribute.

Table 2: Experimental results of the link prediction with Hits@20 performance (top) and the node classification with MAD score (bottom) on benchmark datasets. The extended results for various node embedding methods including SAGE, PNA, and Graphormer are reported in the Appendix.

| | COMPUTERS | PHOTO | CORA ML | OGB-DDI$_{\text{SUBSET}}$ | OGB-DDI$_{\text{FULL}}$ |
|---|---|---|---|---|---|
| *GCN* | 31.01 $\pm$3.37 | 51.05 $\pm$5.45 | 75.93 $\pm$4.36 | 76.11 $\pm$5.92 | 68.18 $\pm$9.24 |
| *NATR$_{GCN}$* | **42.38** $\pm$3.21 | **58.12** $\pm$4.18 | **77.04** $\pm$2.61 | **78.51** $\pm$4.03 | **73.07** $\pm$8.16 |
| *GAT* | 24.73 $\pm$4.96 | 48.23 $\pm$7.43 | 72.42 $\pm$3.45 | 61.46 $\pm$11.51 | 29.02 $\pm$12.52 |
| *NATR$_{GAT}$* | **40.63** $\pm$3.97 | **56.06** $\pm$3.54 | **74.10** $\pm$3.22 | **80.68** $\pm$2.32 | **77.80** $\pm$6.79 |
| *SGC* | 30.37 $\pm$2.73 | 51.31 $\pm$4.80 | 74.49 $\pm$3.03 | 41.04 $\pm$7.12 | 39.19 $\pm$7.87 |
| *NATR$_{SGC}$* | **36.99** $\pm$3.34 | **57.42** $\pm$4.38 | **77.20** $\pm$2.85 | **86.79** $\pm$3.66 | **76.99** $\pm$10.91 |

| | COMPUTERS | | PHOTO | | CORA ML | |
|---|---|---|---|---|---|---|
| | ACCURACY | MAD | ACCURACY | MAD | ACCURACY | MAD |
| *GCN* | 80.12 $\pm$1.71 | 0.46 $\pm$0.03 | 88.50 $\pm$2.11 | 0.83 $\pm$0.06 | 78.71 $\pm$2.00 | 0.55 $\pm$0.03 |
| *NATR$_{GCN}$* | **81.70** $\pm$2.75 | **0.82** $\pm$0.04 | **90.84** $\pm$1.26 | **0.91** $\pm$0.02 | **80.39** $\pm$2.28 | **0.68** $\pm$0.03 |
| *GAT* | 80.86 $\pm$1.95 | 0.63 $\pm$0.05 | 88.87 $\pm$2.04 | 0.57 $\pm$0.04 | 77.35 $\pm$2.02 | **0.84** $\pm$0.04 |
| *NATR$_{GAT}$* | **81.39** $\pm$2.12 | **0.67** $\pm$0.03 | **89.23** $\pm$1.93 | **0.89** $\pm$0.02 | **79.36** $\pm$1.66 | 0.74 $\pm$0.02 |
| *SGC* | 80.31 $\pm$1.53 | 0.26 $\pm$0.03 | 89.18 $\pm$1.67 | 0.45 $\pm$0.07 | 79.30 $\pm$1.89 | 0.34 $\pm$0.02 |
| *NATR$_{SGC}$* | **80.63** $\pm$2.30 | **0.68** $\pm$0.03 | **89.60** $\pm$1.74 | **0.78** $\pm$0.05 | **80.22** $\pm$1.03 | **0.92** $\pm$0.02 |

## 5.2 TASKS

**Link Prediction.** We conduct intensive experiments on the task of link prediction. The attribute-level representation is especially important in predicting potential links between nodes (Li et al., 2018a; Hao et al., 2021). In the case of the OGB-DDI$_{\text{FULL}}$ dataset, the attribute indicates substructures of chemical compounds so it can provide information about potential interactions between drugs in a biological system. The overall performance is reported in Table 2 (top), and the performance based on the number of layers is reported in Table 3.

Table 3: Hits@20 performance on Computers dataset by the number of layers.

| | 2 Layers | 3 Layers | 4 Layers | 5 Layers |
|---|---|---|---|---|
| *GCN* | **31.01** | 30.84 | 28.97 | 26.99 |
| *GCN$_{JK}$* | **29.47** | 27.85 | 28.00 | 27.49 |
| *NATR$_{GCN}$* | 39.81 | 41.54 | 40.96 | **42.38** |
| *GAT* | **24.73** | 21.07 | 11.52 | 4.15 |
| *GAT$_{JK}$* | **27.22** | 24.54 | 23.90 | 23.98 |
| *NATR$_{GAT}$* | 39.51 | 39.58 | **40.63** | 40.21 |
| *SGC* | **30.37** | 25.78 | 24.30 | 23.87 |
| *NATR$_{SGC}$* | **36.99** | 36.47 | 35.31 | 34.01 |

**Node Classification.** Despite the potential benefits of smoothing node representations to be more similar to their neighboring nodes in the node classification task for homogeneous graphs, as opposed to preserving the individual features of each node, our experimental results demonstrate that *NATR* does not impede performance. We also measure the smoothness of node representations based on Mean Average Distance (MAD) (Chen et al., 2020a). The experimental results in the Table 2 (bottom) show that the *NATR* architecture is beneficial in addressing the over-smoothing issue by preserving the individual representation of each node.

## 6 ANALYSIS

### 6.1 IMPROVEMENTS IN THE DILUTION FACTORS OF NATR

**The intra-node dilution factor.** Unlike the $1/|\mathcal{T}_v|$ approach used in MPNNs, *NATR* can enhance the representation of important attributes while suppressing others. The intra-node dilution factor for attribute $t$ is calculated as $\exp\left(Q_v K_t^\top\right)/\sum_{s\in\mathcal{T}_v}\exp\left(Q_v K_s^\top\right)$, which is the attention coefficient at node $v$. In comparison to *GCN*, *NATR$_{GCN}$* increases $\delta_v^{\text{intra}}(t)$ in 38.07% of all cases with a median increase of +30.31% and a maximum increase of +4005.60% on the Computers dataset. The detailed statistics are reported in the Appendix.

**The inter-node dilution factor.** The final representation of node $v$ at the last layer $\tilde{H}_v^{(M)}$ is calculated based on two node-level representations: $H_v^{(M)}$ and $O_v^{(M)}$. The term $H_v^{(M)}$ contains information from its neighboring nodes, while $O_v^{(M)}$ pertains exclusively to node $v$. The inter-node dilution factor

of *NATR* is defined as:

$$\delta^{\text{inter}}(v) = e^T \left[ \frac{\partial \tilde{H}_v^{(M)}}{\partial H_v^{(0)}} + \sum_{m=1}^{M} \frac{\partial \tilde{H}_v^{(M)}}{\partial O_v^{(m)}} \right] e \Big/ e^T \left[ \sum_{u \in \mathcal{V}} \frac{\partial \tilde{H}_v^{(M)}}{\partial H_u^{(0)}} + \sum_{m=1}^{M} \frac{\partial \tilde{H}_v^{(M)}}{\partial O_v^{(m)}} \right] e \quad (11)$$

Even when $\frac{\partial \tilde{H}_v^{(M)}}{\partial H_v^{(0)}}$ value of the numerator is smaller than $\sum_{u \in \mathcal{V}} \frac{\partial \tilde{H}_v^{(M)}}{\partial H_u^{(0)}}$ value of the denominator as in MPNNs, the factor value can still be high in *NATR*. This is primarily due to the contribution from $\sum_{m=1}^{M} \frac{\partial \tilde{H}_v^{(M)}}{\partial O_v^{(m)}}$ helping each node preserve its own feature as shown in Figure 2(b). As demonstrated in Table 3, the performance of MPNNs deteriorates as the depth of the layers increases, whereas *NATR* models exhibit performance gains. In the case of $NATR_{SGC}$, because we adapt the plug-in version that uses over-diluted representations as queries, the performance is slightly decreased. MPNNs with jumping knowledge (JK) (Xu et al., 2018), which concatenate the outputs of all layers, alleviate the performance drop compared to original models. However, JK models fail to improve performance implying that they are inadequate in utilizing the information as the number of layers increases.

## 6.2 EFFECTIVENESS OF NATR

To explore the effectiveness of *NATR*, we measure performance on subsets standing for over-diluted nodes $\mathcal{V}_{Q1}$ and less-diluted nodes $\mathcal{V}_{Q4}$ after two hops aggregation, which are defined as:

$$\mathcal{V}_{Q1} = \{ v \in \mathcal{V} \mid \delta_{\text{Agg}}^{\text{inter}}(v) \leq Q1 \} \quad (12)$$

$$\mathcal{V}_{Q4} = \{ v \in \mathcal{V} \mid \delta_{\text{Agg}}^{\text{inter}}(v) \geq Q3 \ \wedge \ \delta_{\text{Agg}}^{\text{inter}}(v) \neq 1 \}$$

Table 4: Hits@5 performance on subsets of Computers dataset.

|  | $\mathcal{E}_{Q1}$ | $\mathcal{E}_{Q4}$ |
|---|---|---|
| *GCN* | 19.96 | 42.69 |
| $NATR_{GCN}$ | 23.96 (+20.04%) | 45.18 (+5.84%) |
| *GAT* | 13.57 | 34.86 |
| $NATR_{GAT}$ | 24.46 (+80.38%) | 43.49 (+24.74%) |
| *SGC* | 19.39 | 38.61 |
| $NATR_{SGC}$ | 24.10 (+24.29%) | 39.72 (+2.89%) |

where $Q1$ and $Q_3$ represent the first and the third quartiles, which divide the set into the bottom 25% and the top 25% of $\delta_{\text{Agg}}^{\text{inter}}(v)$ values, respectively. We define two subsets of corresponding edges as $\mathcal{E}_{Q1} = \{ (i,j) \in \mathcal{E} \mid i \in \mathcal{V}_{Q1} \vee j \in \mathcal{V}_{Q1} \}$ and $\mathcal{E}_{Q4} = \{ (i,j) \in \mathcal{E} \mid i \in \mathcal{V}_{Q4} \vee j \in \mathcal{V}_{Q4} \}$. The isolated nodes, defined as those with $\delta_{\text{Agg}}^{\text{inter}}(v) = 1$, are excluded. As shown in Table 4, *NATR* models demonstrate improved performance on both subsets, with particularly noteworthy improvement on over-diluted nodes compared to MPNNs.

Furthermore, we compare models with various conditions as described in Table 5. The distinction between (A) and (D) lies in the weight assigned to attribute-level representations. Both are *MLP*-based models but (D) alleviates the intra-node dilution by mixing attributes according to the attention coefficients. The comparison with (D) and (E) shows the effectiveness of the correlation between attributes through $\mathrm{SelfAttn}$ in the attribute encoder. The *GCN* models, (B) and (C), improve performance compared to (A) as a result of incorporating contextual information from neighboring nodes. The complete model (F) exploits *GCN* as a node embedding module, allowing attribute representation to be fused while taking into account the context of the graph.

Table 5: Comparison with various models: (A)-*MLP*, (B)-*GCN*, (C)-$GCN_{JK}$, (D)-$NATR_{MLP}$ with a *MLP* encoder, (E)-$NATR_{MLP}$ with a $\mathrm{SelfAttn}$ encoder, (F)-$NATR_{GCN}$. The model utilizing the correlation between attributes is indicated by CORR, MP denotes the use of message passing, and ATTN indicates that the value is determined by the attention mechanism.

|  | $\delta_v^{intra}(t)$ | $\delta_{\text{Agg}}^{inter}(v)$ | Corr | MP | Hits@20 |
|---|---|---|---|---|---|
| (A) | $1/|\mathcal{T}_v|$ | high | ✗ | ✗ | 20.37 |
| (B) | $1/|\mathcal{T}_v|$ | low | ✗ | ✓ | 31.01 |
| (C) | $1/|\mathcal{T}_v|$ | high | ✗ | ✓ | 29.47 |
| (D) | Attn | high | ✗ | ✗ | 33.26 |
| (E) | Attn | high | ✓ | ✗ | 34.89 |
| (F) | Attn | high | ✓ | ✓ | 42.38 |

## 7 CONCLUSION

In this work, we first introduce the concept of *over-dilution* phenomenon to comprehend the limitations of MPNNs. To assess the over-dilution effect in formal way, we define factors for two sub-phenomena: *intra-node dilution* and *inter-node dilution*. The concept of over-dilution encompasses the diminution of information at both the attribute-level and the node-level. Based on our analysis of the dilution effect, we propose the Node Attribute Transformer (*NATR*) as a solution to alleviate over-dilution and enhance performance. Our approach presents a novel perspective for understanding the limitations of MPNNs and a foundation to the development of more informative representations on graphs.

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

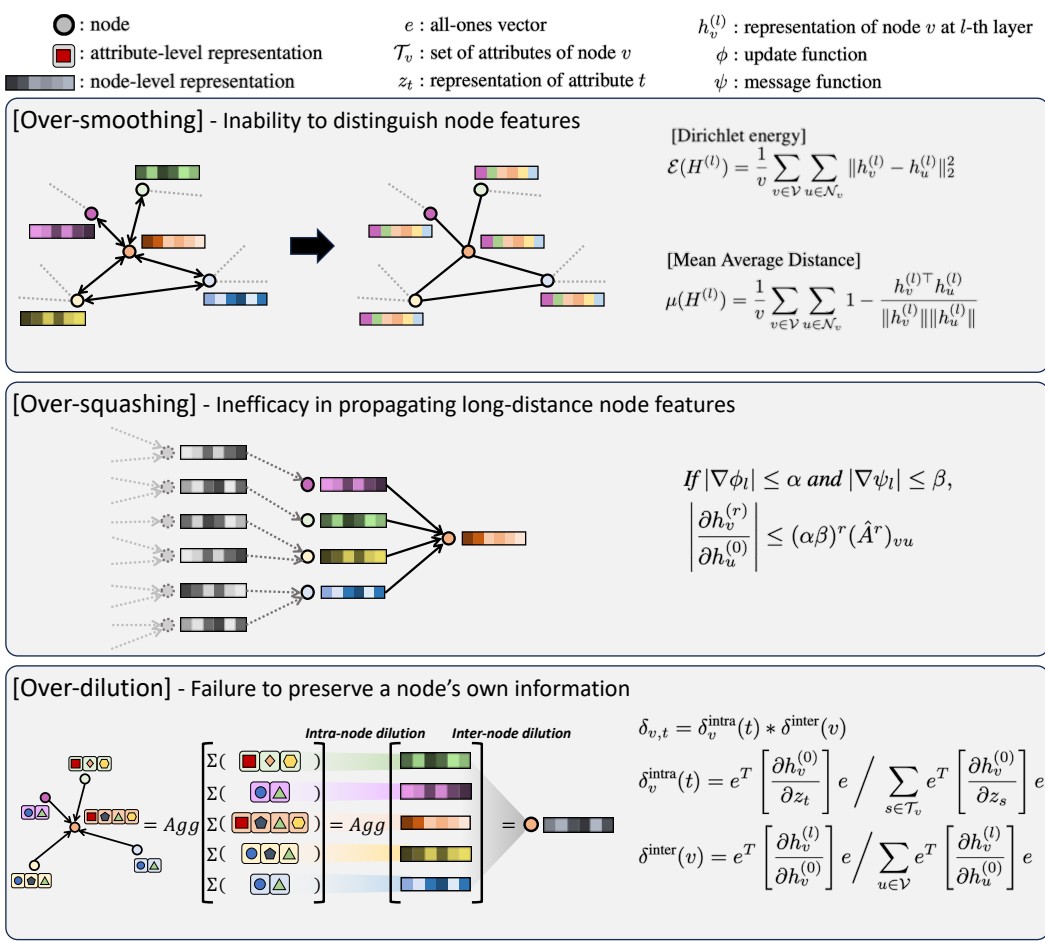

Figure 4: Comparison between concepts: Over-smoothing, Over-squashing, and Over-dilution.

## A  OVER-SMOOTHING AND OVER-SQUASHING

Over-smoothing and over-squashing are both related to over-dilution in that they both pertain to the distortion of information resulting from the irregular structure of graphs. The over-dilution phenomenon is a comprehensive concept that encompasses not only the node-level representation, such as over-smoothing and over-squashing, but also the attribute-level representation, which is essential for achieving informative representations on graphs. Therefore, regardless of the circumstances that trigger over-smoothing and over-squashing, over-dilution can occur at the intra-node level when the number of attributes contained by the nodes becomes excessively large.

**Comparison with over-smoothing**

Over-smoothing refers to the occurrence of similar representations among the nodes after a few layers of message passing, akin to the effect of a low-pass filter (Xu et al., 2018; Li et al., 2018b; Nt & Maehara, 2019; Oono & Suzuki, 2020; Chen et al., 2020a). This is caused by the exchange of information among nodes, which is more likely to occur when nodes have large receptive fields in multi-hop layers. In contrast, over-dilution can occur even with a single aggregation layer as described in the Hypothesis 2 of the main text. As shown in the histogram of $\delta_{\text{Agg}}^{\text{inter}}(v)$ for one hop aggregation in Figure 2(a) of the main text, some nodes already have low $\delta_{\text{Agg}}^{\text{inter}}(v)$ values. Apart from this comparison of two concepts, *NATR* alleviates the over-smoothing issue as shown in the Table 3 of the main text.

**Comparison with over-squashing**

Both phenomena are related in that they may result from the concentration of information from a large number of nodes. However, Over-squashing refers to the propagation of long-range information from node $u$ to node $v$, while Over-dilution refers to the attenuation of information for individual nodes. Over-dilution is distinct from Over-squashing in that it can occur even in the first layer (i.e. with a small receptive field) according to the aggregation coefficients. Specifically, Over-dilution at node $v$ can occur when $\alpha_{vv} \ll \sum_{u \in \tilde{\mathcal{N}}(v) \setminus \{v\}} \alpha_{vu}$, where $\alpha_{vu}$ is defined according to the topology for GCN ($\frac{1}{\sqrt{\deg(v)\deg(u)}}$) and to the attention coefficient for GAT (softmax($\frac{\exp(a_{vu})}{\sum_{w \in \tilde{N}(v)} \exp(a_{vw})}$)). It is true that Over-dilution occurs more often as the range increases, but as can be seen in the histogram of $\delta_{\text{Agg}}^{\text{inter}}(v)$, it also occurs frequently in short-range.

# B  INTRA-DILUTION FACTOR

Unlike MPNNs, which assign equal weight (i.e. $\frac{1}{|\mathcal{T}_v|}$) to all attributes during the construction of the node representation, *NATR* merges attribute representations based on the attention coefficients between node and attribute representations (i.e. $\exp(Q_v K_t^\top) / \sum_{s \in \mathcal{T}_v} \exp(Q_v K_s^\top)$). To examine the ability of *NATR* to prevent over-dilution at the intra-node level, we construct synthetic dataset. While keeping the original graph topology (i.e. adjacency matrix) of the CoraML dataset, we generate a new attribute set and assign 10 attributes to each class as key attributes. The key attributes assigned to each class are highly likely to be held by the node (e.g. with $p = 0.8$), while the remaining non-key attributes are less likely to be held (e.g. with $p = 0.2$). The train, validation, test sets include 1000, 210, and 1785 nodes, respectively. Overall, for all nodes in the synthetic data's test set, $NATR_{GCN}$ amplifies the influence of key attributes in the node representation by approximately 1508.62% in comparison to non-key attributes. This approach effectively curbs over-dilution of key attributes at the intra-node level. We randomly sample 10 nodes for each class in the test set and visualize the intra-node dilution factors of $NATR_{GCN}$ in the Figure 5.

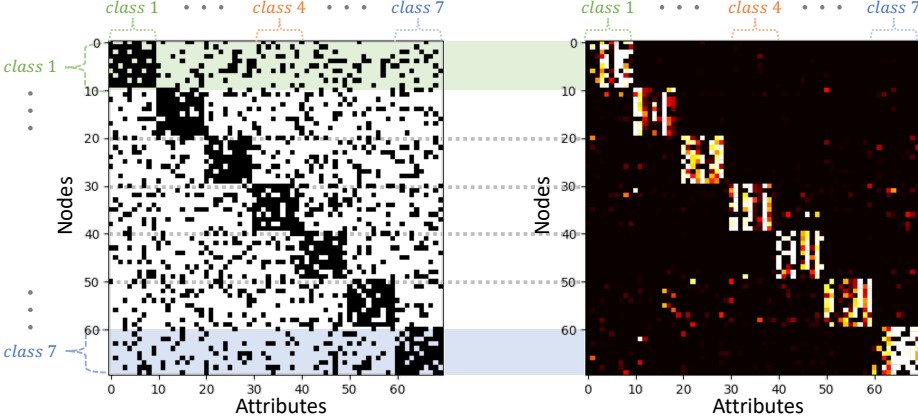

Figure 5: This figure presents a visualization of the synthetic node feature matrix (left) and the intra-node dilution factors of $NATR_{GCN}$ (right). The $y$-axis represents nodes sorted according to the class and the $x$-axis represents attributes. We randomly sample 10 nodes for each class in the test set for this visualization. We assign the 10 attributes to each class sequentially, identifying them as the block-diagonal elements in the matrix and principal attributes for class prediction. The remaining attributes are considered noise. As shown in the figure on the right, $NATR_{GCN}$ amplifies the attribute representations pertinent to each node's class, while diminishing the influence of other attributes. On average, across all nodes in the synthetic data test set, $NATR_{GCN}$ boosts the strength of the key attributes in the node representation by 1508.62% compared to non-key attributes, thus preventing their over-dilution at the intra-node level.

We define a new quantitative metric $\bar{\delta}_v^{\text{intra}}(t)$ as below to examine how much *NATR* adjusts the dilution factors compared to the original ones $\hat{\delta}_v^{\text{intra}}(t) = \frac{1}{|\mathcal{T}_v|}$.

$$\bar{\delta}_v^{\text{intra}}(t) = \frac{\delta_v^{\text{intra}}(t) - \hat{\delta}_v^{\text{intra}}(t)}{\hat{\delta}_v^{\text{intra}}(t)} * 100(\%)$$

where $\delta_v^{\text{intra}}(t)$ is the intra-node dilution factor of *NATR*. we consider the Computers dataset (3,675,081 instances of the intra-node dilution factor). $NATR_{GCN}$ enhances the importance of 38.07% cases ($\bar{\delta}_v^{\text{intra}}(t) > 0$) compared to $GCN$, with a median gain +30.31% and a maximum gain +4005.60%. $NATR_{GCN}$ suppresses the importance of 61.92% cases ($\bar{\delta}_v^{\text{intra}}(t) < 0$) compared to $GCN$, with median -29.46% and minimum -95.82%. In contrast to MPNNs, where attributes are equally diluted regardless of their importance, NATR is designed to prioritize important attributes ($\bar{\delta}_v^{\text{intra}}(t) > 0$) and give less weight to unimportant ones ($\bar{\delta}_v^{\text{intra}}(t) < 0$). The detailed statistics are reported in Table 6.

Table 6: (left) $NATR_{GCN}$ enhances the importance of attributes $\bar{\delta}_v^{\text{intra}}(t) > 0$. Remarkably, there are instances where the improvement ranges from 2 to 40 times compared to $GCN$, which constitutes 12.53% of the enhanced cases. (right) $NATR_{GCN}$ suppresses the importance of attributes $\bar{\delta}_v^{\text{intra}}(t) \leq 0$.

| $\bar{\delta}_v^{\text{INTRA}}(t) > 0$ | COUNT | PROPORTION || $\bar{\delta}_v^{\text{INTRA}}(t) \leq 0$ | COUNT | PROPORTION |
|---|---|---|---|---|---|
| $0\% \sim +10\%$ | 285,150 | 20.38% | $0\% \sim -10\%$ | 344,872 | 15.15% |
| $+10\% \sim +20\%$ | 229,552 | 16.41% | $-10\% \sim -20\%$ | 395,988 | 17.40% |
| $+20\% \sim +30\%$ | 179,963 | 12.86% | $-20\% \sim -30\%$ | 419,467 | 18.43% |
| $+30\% \sim +40\%$ | 139,937 | 10.00% | $-30\% \sim -40\%$ | 398,607 | 17.51% |
| $+40\% \sim +50\%$ | 108,893 | 7.78% | $-40\% \sim -50\%$ | 327,231 | 14.38% |
| $+50\% \sim +60\%$ | 852,85 | 6.10% | $-50\% \sim -60\%$ | 222,725 | 9.79% |
| $+60\% \sim +70\%$ | 67,096 | 4.80% | $-60\% \sim -70\%$ | 118,015 | 5.19% |
| $+70\% \sim +80\%$ | 53,009 | 3.79% | $-70\% \sim -80\%$ | 41,757 | 1.83% |
| $+80\% \sim +90\%$ | 41,827 | 2.99% | $-80\% \sim -90\%$ | 6,988 | 0.31% |
| $+90\% \sim +100\%$ | 33,254 | 2.38% | $-90\% \sim -100\%$ | 185 | 0.01% |
| $+100\% \sim +4006\%$ | 175,280 | 12.53% | | | |
| | | 100.0% || | | 100.0% |

## C  INTER-DILUTION FACTOR

As described in Eq. (4) of the main text, the inter-node dilution factor is defined as:

$$\delta^{\text{inter}}(v) = \frac{e^T \left[ \frac{\partial h_v^{(l)}}{\partial h_v^{(0)}} \right] e}{\sum_{u \in \mathcal{V}} e^T \left[ \frac{\partial h_v^{(l)}}{\partial h_u^{(0)}} \right] e}$$

To observe the effect of the aggregation exclusively, we eliminate the effect of the non-linear transformation by setting all weight and initial node feature matrices to be the identity matrix. The inter-dilution factor (aggregation-only) $\delta_{\text{Agg}}^{\text{inter}}(v)$ is calculated as:

$$\delta_{\text{Agg}}^{\text{inter}}(v) = \frac{e^T \left[ \frac{\partial h_v'^{(l)}}{\partial h_v^{(0)}} \right] e}{\sum_{u \in \mathcal{V}} e^T \left[ \frac{\partial h_v'^{(l)}}{\partial h_u^{(0)}} \right] e}, \quad \text{where} \quad h'^{(l)} = (\tilde{D}^{-\frac{1}{2}} \tilde{A} \tilde{D}^{-\frac{1}{2}})^l I_{N_{\mathcal{V}}} \text{ for } GCN$$

## C.1 THE NUMERATOR OF $\delta_{\text{AGG}}^{\text{INTER}}(v)$

The numerator of $\delta_{\text{Agg}}^{\text{inter}}(v)$ is calculated from:

$$\frac{\partial h_v^{\prime(l)}}{\partial h_v^{(0)}} = \underbrace{\prod_{i=1}^{l} \alpha_{vv}^{(i)} \cdot \frac{\partial h_v^{(0)}}{\partial h_v^{(0)}}}_{\text{for } l \geq 1} + \underbrace{\sum_{u \in \tilde{\mathcal{N}}(v) \backslash \{v\}} \sum_{k=1}^{l-1} \left( \prod_{\substack{j=k+2 \\ k \leq l-2}}^{l} \alpha_{vv}^{(j)} \right) \alpha_{vu}^{(k+1)} \frac{\partial h_u^{\prime(k)}}{\partial h_v^{(0)}}}_{\text{for } l \geq 2}$$

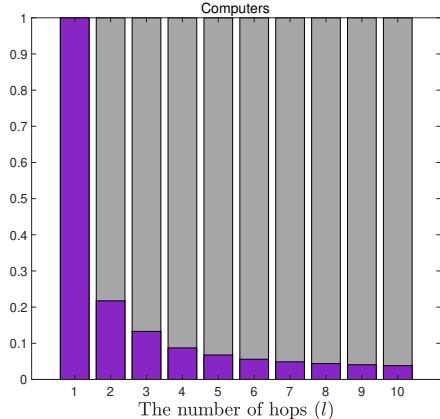

Figure 6: The purple bar represents the average ratio of the former term $e^T \left[ \left( \frac{1}{\deg(v)} \right)^l \frac{\partial h_v^{(0)}}{\partial h_v^{(0)}} \right] e$, which is the amount (influence) of the initial node-level feature $h_v^{(0)}$ that is not propagated to other nodes. The gray bar represents the average ratio of the latter term $e^T \left[ \sum_{k=1}^{l-1} \sum_{u \in \tilde{\mathcal{N}}(v) \backslash \{v\}} \frac{1}{\sqrt{\deg(v)} \sqrt{\deg(u)}} \left( \frac{1}{\deg(v)} \right)^{l-k-1} \frac{\partial h_u^{\prime(k)}}{\partial h_v^{(0)}} \right] e$, which is the amount (influence) of the representation preserved by its neighbors. It implies that in the *GCN*, the representation of each node is primarily maintained by its neighboring nodes, with the representation being 'received back' from them after two hops aggregation.

For $l = 1$,

$$\frac{\partial h_v^{\prime(1)}}{\partial h_v^{(0)}} = \alpha_{vv}^{(1)} \frac{\partial h_v^{(0)}}{\partial h_v^{(0)}}$$

For $l = 2$,

$$\frac{\partial h_v^{\prime(2)}}{\partial h_v^{(0)}} = \alpha_{vv}^{(1)} \alpha_{vv}^{(2)} \frac{\partial h_v^{(0)}}{\partial h_v^{(0)}} + \sum_{u \in \tilde{N}(v) \backslash \{v\}} \alpha_{vu}^{(2)} \frac{\partial h_u^{\prime(1)}}{\partial h_v^{(0)}}$$

For $l = 3$,

$$\frac{\partial h_v^{\prime(3)}}{\partial h_v^{(0)}} = \alpha_{vv}^{(3)} \alpha_{vv}^{(2)} \alpha_{vv}^{(1)} \frac{\partial h_v^{(0)}}{\partial h_v^{(0)}} + \sum_{u \in \tilde{N}(v) \backslash \{v\}} \left( \alpha_{vv}^{(3)} \alpha_{vu}^{(2)} \frac{\partial h_u^{\prime(1)}}{\partial h_v^{(0)}} + \alpha_{vu}^{(3)} \frac{\partial h_u^{\prime(2)}}{\partial h_v^{(0)}} \right)$$

For $l = 4$,

$$\frac{\partial h_v^{\prime(4)}}{\partial h_v^{(0)}} = \alpha_{vv}^{(4)} \alpha_{vv}^{(3)} \alpha_{vv}^{(2)} \alpha_{vv}^{(1)} \frac{\partial h_v^{(0)}}{\partial h_v^{(0)}} + \sum_{u \in \tilde{N}(v) \backslash \{v\}} \left( \alpha_{vv}^{(4)} \alpha_{vv}^{(3)} \alpha_{vu}^{(2)} \frac{\partial h_u^{\prime(1)}}{\partial h_v^{(0)}} + \alpha_{vv}^{(4)} \alpha_{vu}^{(3)} \frac{\partial h_u^{\prime(2)}}{\partial h_v^{(0)}} + \alpha_{vu}^{(4)} \frac{\partial h_u^{\prime(3)}}{\partial h_v^{(0)}} \right)$$

For *GCN*,

$$
\begin{aligned}
\frac{\partial h_v'^{(l)}}{\partial h_v^{(0)}} &= \frac{1}{\sqrt{\deg(v)}} \cdot \operatorname{diag}\left(\mathbb{1}_{f_v^{(l)}>0}\right) \cdot W^{(l)} \cdot \sum_{u \in \check{\mathcal{N}}(v)} \frac{1}{\sqrt{\deg(u)}} \frac{\partial h_u^{(l-1)}}{\partial h_v^{(0)}} \\
&= \frac{1}{\sqrt{\deg(v)}} \cdot \sum_{u \in \check{\mathcal{N}}(v)} \frac{1}{\sqrt{\deg(u)}} \frac{\partial h_u'^{(l-1)}}{\partial h_v^{(0)}} \\
&= \frac{1}{\deg(v)} \frac{\partial h_v'^{(l-1)}}{\partial h_v^{(0)}} + \sum_{u \in \check{\mathcal{N}}(v)\setminus\{v\}} \frac{1}{\sqrt{\deg(v)}\sqrt{\deg(u)}} \frac{\partial h_u'^{(l-1)}}{\partial h_v^{(0)}} \\
&= \frac{1}{\deg(v)} \left( \frac{1}{\deg(v)} \frac{\partial h_v'^{(l-2)}}{\partial h_v^{(0)}} + \sum_{u \in \check{\mathcal{N}}(v)\setminus\{v\}} \frac{1}{\sqrt{\deg(v)}\sqrt{\deg(u)}} \frac{\partial h_u'^{(l-2)}}{\partial h_v^{(0)}} \right) \\
&\quad + \sum_{u \in \check{\mathcal{N}}(v)\setminus\{v\}} \frac{1}{\sqrt{\deg(v)}\sqrt{\deg(u)}} \frac{\partial h_u'^{(l-1)}}{\partial h_v^{(0)}} \\
&= \left(\frac{1}{\deg(v)}\right)^l \frac{\partial h_v^{(0)}}{\partial h_v^{(0)}} + \sum_{k=1}^{l-1} \sum_{u \in \check{\mathcal{N}}(v)\setminus\{v\}} \frac{1}{\sqrt{\deg(v)}\sqrt{\deg(u)}} \left(\frac{1}{\deg(v)}\right)^{l-k-1} \frac{\partial h_u'^{(k)}}{\partial h_v^{(0)}}
\end{aligned}
$$

where $f_v^{(l)}$ represents the pre-activated feature of $h_v^{(l)}$ and $k, j \in \mathbb{N}$.

We can interpret the former term $\prod_{i=1}^{l} \alpha_{vv}^{(i)} \cdot \frac{\partial h_v^{(0)}}{\partial h_v^{(0)}}$ as the amount (influence) of the initial node-level feature $h_v^{(0)}$ that is not propagated to other nodes.

For the latter term $\sum_{u \in \check{\mathcal{N}}(v)\setminus\{v\}} \sum_{k=1}^{l-1} \left( \prod_{\substack{j=k+2 \\ k \le l-2}}^{l} \alpha_{vv}^{(j)} \right) \alpha_{vu}^{(k+1)} \frac{\partial h_u^{(k)}}{\partial h_v^{(0)}}$, $\frac{\partial h_u^{(k)}}{\partial h_v^{(0)}}$ is the amount (influence) of the representation of node $v$ that node $u$ has at $k$-th layer. It is passed to node $v$ at the next layer, which is $(k+1)$-th layer, with aggregation coefficient $\alpha_{vu}^{(k+1)}$. Then it is regarded as a portion of own representation $\frac{\partial h_v^{(k+1)}}{\partial h_v^{(0)}}$ and diluted with a factor $\prod_{\substack{j=k+2 \\ k \le l-2}}^{l} \alpha_{vv}^{(j)}$, which is the product of aggregation coefficients for self-loop from $(k+2)$-th layer to $(l)$-th layer.

Figure 6 shows the ratio between the former term and the latter term.

## C.2 THE DENOMINATOR OF $\delta_{\text{AGG}}^{\text{INTER}}(v)$

The denominator of $\delta_{\text{Agg}}^{\text{inter}}(v)$ is calculated from:

$$
\sum_{u \in \mathcal{V}} \frac{\partial h_v'^{(l)}}{\partial h_u^{(0)}} = \sum_{x \in \check{N}(v)} \sum_{u \in \mathcal{V}} \sum_{k=0}^{l-1} \left( \prod_{\substack{j=k+2 \\ k \le l-2}}^{l} \alpha_{vv}^{(j)} \right) \alpha_{vx}^{(k+1)} \frac{\partial h_x'^{(k)}}{\partial h_u^{(0)}}
$$

For $l = 1$,

$$
\sum_{u \in \mathcal{V}} \frac{\partial h_v'^{(1)}}{\partial h_u^{(0)}} = \sum_{u \in \check{N}(v)} \alpha_{vu}^{(1)} \frac{\partial h_u^{(0)}}{\partial h_u^{(0)}}
$$

For $l = 2$,

$$
\sum_{u \in \mathcal{V}} \frac{\partial h_v'^{(2)}}{\partial h_u^{(0)}} = \sum_{x \in \check{N}(v)} \left( \alpha_{vv}^{(2)} \alpha_{vx}^{(1)} \frac{\partial h_x^{(0)}}{\partial h_x^{(0)}} + \sum_{u \in \mathcal{V}} \alpha_{vx}^{(2)} \frac{\partial h_x'^{(1)}}{\partial h_u^{(0)}} \right)
$$

For $l = 3$,

$$\sum_{u \in \mathcal{V}} \frac{\partial h_v^{'(3)}}{\partial h_u^{(0)}} = \sum_{x \in \tilde{N}(v)} \left( \alpha_{vv}^{(3)} \alpha_{vv}^{(2)} \alpha_{vx}^{(1)} \frac{\partial h_x^{(0)}}{\partial h_x^{(0)}} + \sum_{u \in \mathcal{V}} \alpha_{vv}^{(3)} \alpha_{vx}^{(2)} \frac{\partial h_x^{'(1)}}{\partial h_u^{(0)}} + \sum_{u \in \mathcal{V}} \alpha_{vx}^{(3)} \frac{\partial h_x^{'(2)}}{\partial h_u^{(0)}} \right)$$

$$= \sum_{x \in \tilde{N}(v)} \sum_{u \in \mathcal{V}} \left( \alpha_{vv}^{(3)} \alpha_{vv}^{(2)} \alpha_{vx}^{(1)} \frac{\partial h_x^{(0)}}{\partial h_u^{(0)}} + \alpha_{vv}^{(3)} \alpha_{vx}^{(2)} \frac{\partial h_x^{'(1)}}{\partial h_u^{(0)}} + \alpha_{vx}^{(3)} \frac{\partial h_x^{'(2)}}{\partial h_u^{(0)}} \right)$$

For $l = 4$,

$$\sum_{u \in \mathcal{V}} \frac{\partial h_v^{'(4)}}{\partial h_u^{(0)}} = \sum_{x \in \tilde{N}(v)} \sum_{u \in \mathcal{V}} \left( \alpha_{vv}^{(4)} \alpha_{vv}^{(3)} \alpha_{vv}^{(2)} \alpha_{vx}^{(1)} \frac{\partial h_x^{(0)}}{\partial h_u^{(0)}} + \alpha_{vv}^{(4)} \alpha_{vv}^{(3)} \alpha_{vx}^{(2)} \frac{\partial h_x^{'(1)}}{\partial h_u^{(0)}} + \alpha_{vv}^{(4)} \alpha_{vx}^{(3)} \frac{\partial h_x^{'(2)}}{\partial h_u^{(0)}} + \alpha_{vx}^{(4)} \frac{\partial h_x^{'(3)}}{\partial h_u^{(0)}} \right)$$

For *GCN*,

$$\sum_{u \in \mathcal{V}} \frac{\partial h_v^{'(l)}}{\partial h_u^{(0)}} = \sum_{x \in \tilde{N}(v)} \sum_{u \in \mathcal{V}} \sum_{k=0}^{l-1} \left( \frac{1}{\deg(v)} \right)^{l-k-1} \frac{1}{\sqrt{\deg(v)}\sqrt{\deg(u)}} \frac{\partial h_x^{'(k)}}{\partial h_u^{(0)}}$$

## C.3 THE FINAL FORM OF $\delta_{\text{AGG}}^{\text{INTER}}(v)$

The final form of $\delta_{\text{Agg}}^{\text{inter}}(v)$ is defined as:

$$\delta_{\text{Agg}}^{\text{inter}}(v) = \frac{e^T \left[ \prod_{i=1}^{l} \alpha_{vv}^{(i)} \frac{\partial h_v^{(0)}}{\partial h_v^{(0)}} + \sum_{u \in \tilde{\mathcal{N}}(v) \setminus \{v\}} \sum_{k=1}^{l-1} \left( \prod_{\substack{j=k+2 \\ k \leq l-2}}^{l} \alpha_{vv}^{(j)} \right) \alpha_{vu}^{(k+1)} \frac{\partial h_u^{'(k)}}{\partial h_v^{(0)}} \right] e}{\sum_{u \in \mathcal{V}} e^T \left[ \sum_{x \in \tilde{N}(v)} \sum_{u \in \mathcal{V}} \sum_{k=0}^{l-1} \left( \prod_{\substack{j=k+2 \\ k \leq l-2}}^{l} \alpha_{vv}^{(j)} \right) \alpha_{vx}^{(k+1)} \frac{\partial h_x^{'(k)}}{\partial h_u^{(0)}} \right] e}$$

For $l = 1$ (used in Hypothesis 2 of the main text),

$$\delta_{\text{Agg}}^{\text{inter}}(v) = \frac{e^T \left[ \alpha_{vv}^{(1)} \frac{\partial h_v^{(0)}}{\partial h_v^{(0)}} \right] e}{e^T \left[ \sum_{u \in \tilde{N}(v)} \alpha_{vu}^{(1)} \frac{\partial h_u^{(0)}}{\partial h_u^{(0)}} \right] e} = \frac{\alpha_{vv}^{(1)}}{\sum_{u \in \tilde{\mathcal{N}}(v)} \alpha_{vu}^{(1)}}$$

For $l = 2$,

$$\delta_{\text{Agg}}^{\text{inter}}(v) = \frac{e^T \left[ \alpha_{vv}^{(1)} \alpha_{vv}^{(2)} \cdot \frac{\partial h_v^{(0)}}{\partial h_v^{(0)}} + \sum_{u \in \tilde{\mathcal{N}}(v) \setminus \{v\}} \alpha_{vu}^{(2)} \frac{\partial h_u^{'(1)}}{\partial h_v^{(0)}} \right] e}{e^T \left[ \sum_{x \in \tilde{N}(v)} \sum_{u \in \mathcal{V}} \left( \alpha_{vv}^{(2)} \alpha_{vx}^{(1)} \frac{\partial h_x^{(0)}}{\partial h_u^{(0)}} + \sum_{u \in \mathcal{V}} \alpha_{vx}^{(2)} \frac{\partial h_x^{'(1)}}{\partial h_u^{(0)}} \right) \right] e}$$

Figure 7 shows that the inter-dilution factor (aggregation-only) is dramatically decreased as the number of layers and the size of the receptive field increase for all datasets. The histograms of the inter-dilution factor (aggregation-only) and the average size of the receptive field are shown in Figure 8.

For *NATR*,

$$\delta_{\text{Agg}}^{\text{inter}}(v) = \frac{e^T \left[ \prod_{i=1}^{l} \alpha_{vv}^{(i)} \frac{\partial h_v^{(0)}}{\partial h_v^{(0)}} + \sum_{u \in \tilde{\mathcal{N}}(v) \setminus \{v\}} \sum_{k=1}^{l-1} \left( \prod_{\substack{j=k+2 \\ k \leq l-2}}^{l} \alpha_{vv}^{(j)} \right) \alpha_{vu}^{(k+1)} \frac{\partial h_u^{'(k)}}{\partial h_v^{(0)}} \right] e + e^T \left[ \sum_{m=1}^{M} \frac{\partial \tilde{H}_v^{(M)}}{\partial O_v^{(m)}} \right] e}{\sum_{u \in \mathcal{V}} e^T \left[ \sum_{x \in \tilde{N}(v)} \sum_{u \in \mathcal{V}} \sum_{k=0}^{l-1} \left( \prod_{\substack{j=k+2 \\ k \leq l-2}}^{l} \alpha_{vv}^{(j)} \right) \alpha_{vx}^{(k+1)} \frac{\partial h_x^{'(k)}}{\partial h_u^{(0)}} \right] e + e^T \left[ \sum_{m=1}^{M} \frac{\partial \tilde{H}_v^{(M)}}{\partial O_v^{(m)}} \right] e}$$

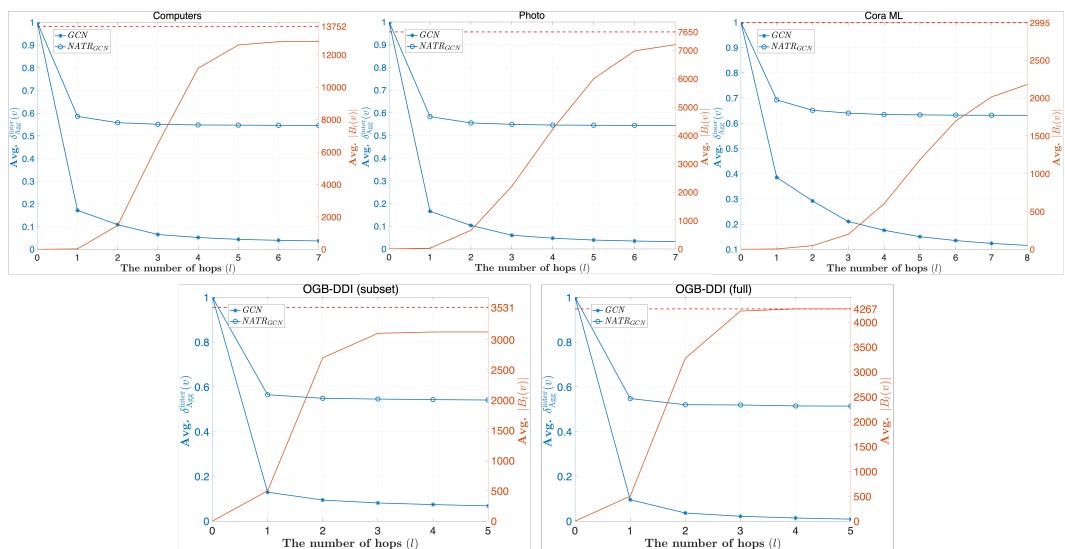

Figure 7: The average of the inter-dilution factor (aggregation-only) with the left $y$-axis (blue line) and the average size of the receptive field (orange line) with the right $y$-axis in all datasets. The $x$-axis represents the number of hops for the aggregation.

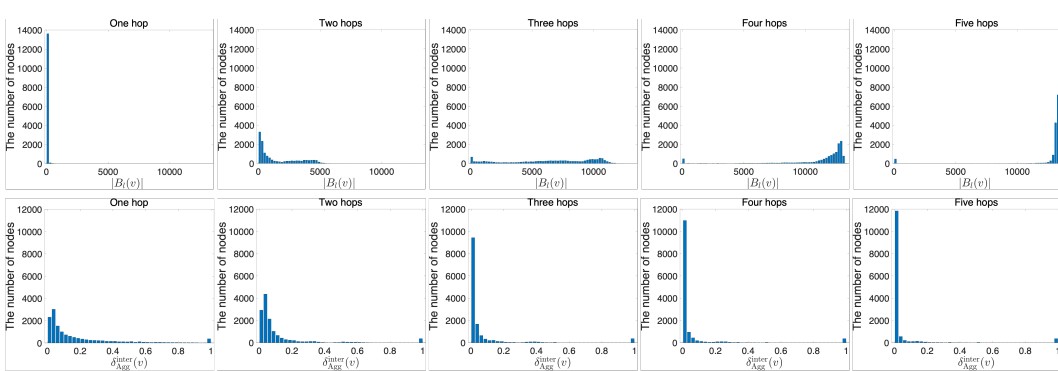

Figure 8: Histograms of the average size of the receptive field (top) and the inter-node dilution factor (aggregation-only) (bottom), from one hop (left) to five hops (right).

For *NATR* with $l = 1$,

$$
\delta_{\text{Agg}}^{\text{inter}}(v) = \frac{e^T \left[ \alpha_{vv}^{(1)} \dfrac{\partial h_v^{(0)}}{\partial h_v^{(0)}} \right] e \; + e^T \left[ \dfrac{\partial \tilde{H}_v^{(1)}}{\partial O_v^{(1)}} \right] e}{e^T \left[ \sum_{u \in \tilde{N}(v)} \alpha_{vu}^{(1)} \dfrac{\partial h_u^{(0)}}{\partial h_u^{(0)}} \right] e + e^T \left[ \dfrac{\partial \tilde{H}_v^{(1)}}{\partial O_v^{(1)}} \right] e}
$$

For *NATR* with $l = 2$,

$$
\delta_{\text{Agg}}^{\text{inter}}(v) = \frac{e^T \left[ \alpha_{vv}^{(1)} \alpha_{vv}^{(2)} \cdot \dfrac{\partial h_v^{(0)}}{\partial h_v^{(0)}} + \sum_{u \in \tilde{N}(v) \backslash \{v\}} \alpha_{vu}^{(2)} \dfrac{\partial h_u'^{(1)}}{\partial h_v^{(0)}} \right] e \; + e^T \left[ \dfrac{\partial \tilde{H}_v^{(2)}}{\partial O_v^{(1)}} + \dfrac{\partial \tilde{H}_v^{(2)}}{\partial O_v^{(2)}} \right] e}{e^T \left[ \sum_{x \in \tilde{N}(v)} \sum_{u \in \mathcal{V}} \left( \alpha_{vv}^{(2)} \alpha_{vx}^{(1)} \dfrac{\partial h_x^{(0)}}{\partial h_u^{(0)}} + \sum_{u \in \mathcal{V}} \alpha_{vx}^{(2)} \dfrac{\partial h_x'^{(1)}}{\partial h_u^{(0)}} \right) \right] e + e^T \left[ \dfrac{\partial \tilde{H}_v^{(2)}}{\partial O_v^{(1)}} + \dfrac{\partial \tilde{H}_v^{(2)}}{\partial O_v^{(2)}} \right] e}
$$

# D  DATASETS

Table 7: Dataset statistics.

|  | $|\mathcal{V}|$ | $|\mathcal{T}|$ | $|\mathcal{E}_{train}|$ | $|\mathcal{E}_{valid}|$ | $|\mathcal{E}_{test}|$ |
|---|---|---|---|---|---|
| AMAZON COMPUTERS | 13752 | 767 | 196689 | 24586 | 24586 |
| AMAZON PHOTO | 7650 | 745 | 95265 | 11908 | 11908 |
| CORA ML | 2995 | 2879 | 6936 | 407 | 815 |
| OGB-DDI$_{SUBSET}$ | 3531 | 1024 | 882012 | 110446 | 114363 |
| OGB-DDI$_{FULL}$ | 4267 | 1024+1 | 1067911 | 133489 | 133489 |

As described in Table 7, we split the edge set into train/valid/test without overlap as 80/10/10 for Computers and Photo (Shchur et al., 2018) and 85/5/10 for Cora ML (Bojchevski & Günnemann, 2018). For OGB-DDI$_{FULL}$ and OGB-DDI$_{SUBSET}$ datasets, we use pre-defined data split (Hu et al., 2020). We randomly sample the negative edges with the same number of the positive set. We assume that the node has a discrete binary vector $X_v \in \mathbb{R}^{N_\mathcal{T}}$, where $X_{v,t} \in \{0,1\}$ in the paper. However, even in the dataset where each node has a continuous vector, *NATR* works just as well. The values in $X_v$ can be interpreted as the pre-defined magnitude of each attribute in the node. We can binarize these values to fit our assumption because the pre-defined magnitude may not match the importance in representation learning. There are several methods for binarization, such as thresholding and discretization into a set of discrete bins. In another way, we can incorporate the magnitude into the attention coefficient of the decoder as $\exp{(Q_v K_t^\top + X_{v,t})}/\sum_{s \in \mathcal{T}_v} \exp{(Q_v K_s^\top + X_{v,s})}$.

# E  DETAILS ON TRAINING

We conduct the grid search for hyperparameter configurations. The search space sets are $\{2, 3, 4, 5\}$ for the number of layers, $\{64, 128, 256\}$ for the hidden dimension, and $\{0.01, 0.005, 0.001, 0.0005\}$ for the learning rate. For *NATR*, $d_{FFN}$ is set to 512 and the number of encoder layers is set to two. We apply the dropout with $p = 0.5$, 10k epochs of the optimization step, and the early stopping with 1k epochs at the hyperparameter search for all models. As done for other transformers, we train *NATR* with the auxiliary loss for the outputs from intermediate layers of the decoder.

# F  COMPATIBILITY WITH OTHER NODE EMBEDDING MODELS

Table 8: Comparison of Hits@20 on the Computer dataset with various node embedding models.

|  | SAGE | GATV2 | ARMA | PNA$_{t=2}$ | PNA$_{t=4}$ | GCNII$_{\alpha=0.1}$ | GCNII$_{\alpha=0.5}$ | GRAPHORMER |
|---|---|---|---|---|---|---|---|---|
| *Node Model* | 32.30 $_{\pm4.32}$ | 28.69 $_{\pm2.46}$ | 31.42 $_{\pm3.07}$ | 21.77 $_{\pm3.82}$ | 29.41 $_{\pm3.56}$ | 30.00 $_{\pm3.69}$ | 27.54 $_{\pm3.27}$ | 25.94 $_{\pm4.32}$ |
| *NATR$_{NodeModel}$* | 36.11 $_{\pm3.75}$ | 37.52 $_{\pm3.37}$ | 37.23 $_{\pm3.91}$ | 31.62 $_{\pm2.66}$ | 33.68 $_{\pm4.03}$ | 37.47 $_{\pm4.56}$ | 36.78 $_{\pm3.03}$ | 30.71 $_{\pm3.68}$ |

The *NATR* architecture can be attached to various node embedding models for improving performance. Any kind of models for learning node-level representations can be exploited as $NodeModule$ in the decoder of *NATR*. We conduct extended comparison with various node embedding models and report the results in Table 8. *SAGE* (Hamilton et al., 2017) aggregates node representations with the coefficient defined as $\alpha_{vv} = \alpha_{vu} = \frac{1}{\deg(v)}$ and *ARMA* (Bianchi et al., 2021) indicates the graph convolution inspired by the auto-regressive moving average filter which is robust to noise and better to capture the global graph structure. *PNA* (Corso et al., 2020) combines multiple aggregators with scalers for degree, which is a generalized sum aggregator. *GCNII* (Chen et al., 2020b) updates node representations as $h^{(l)} = \left( (1 - \alpha^{(l)})\tilde{D}^{-\frac{1}{2}}\tilde{A}\tilde{D}^{-\frac{1}{2}}h^{(l-1)} + \alpha^{(l)}h^{(0)} \right)((1 - \beta^{(l)})I + \beta^{(l)}\Theta)$ while preserving the initial feature $h^{(0)}$. The *NATR* architecture shows performance gains through the integration of various node embedding models.

# G INVESTIGATION OF THE EFFECTIVENESS OF *NATR*, INCLUDING ABLATION STUDIES

Table 9: Ablation studies for *NATR* with the link prediction performance on the Computers dataset.

| | ENCODER | | DECODER | | AUX LOSS | HITS@20 | REMARKS |
|---|---|---|---|---|---|---|---|
| | NUM. LAYERS | TYPE | NUM. LAYERS | NODE MODULE | | | |
| (REF1) | 2 | SelfAttn | 5 | GCN | ✓ | 42.38 ±3.21 | - |
| (REF2) | 2 | SelfAttn | 2 | GCN | ✓ | 39.81 ±2.88 | - |
| (a) | 2 | SelfAttn | 5 | MLP | ✓ | 34.89 ±3.42 | *MLP for Node Embedding (w/o aggregation)* |
| (b) | 2 | MLP | 5 | GCN | ✓ | 42.03 ±3.06 | *MLP for Encoder (w/o correlation of attributes)* |
| (c) | 2 | MLP | 5 | MLP | ✓ | 33.26 ±3.20 | *MLP Node Embedding, MLP Encoder* |
| (d) | 2 | SelfAttn | 5 | GCN | ✗ | 22.50 ±3.32 | *w/o intermediate loss* |
| (e) | 2 | SelfAttn | 2 | GCN | ✗ | 33.04 ±4.09 | *Two decoder layers w/o intermediate loss* |
| (f) | 1 | SelfAttn | 5 | GCN | ✓ | 42.66 ±3.95 | *The number of encoder layers* |
| (g) | 3 | SelfAttn | 5 | GCN | ✓ | 42.62 ±3.55 | *The number of encoder layers* |
| (h) | 4 | SelfAttn | 5 | GCN | ✓ | 43.15 ±2.94 | *The number of encoder layers* |
| (i) | 5 | SelfAttn | 5 | GCN | ✓ | 43.00 ±3.43 | *The number of encoder layers* |
| (j) | 6 | SelfAttn | 5 | GCN | ✓ | 42.34 ±2.87 | *The number of encoder layers* |
| (k) | 2 | SelfAttn | 3 | GCN | ✓ | 41.54 ±3.70 | *The number of decoder layers* |
| (l) | 2 | SelfAttn | 4 | GCN | ✓ | 40.96 ±4.19 | *The number of decoder layers* |
| (m) | 2 | SelfAttn | 6 | GCN | ✓ | 41.06 ±4.36 | *The number of decoder layers* |
| (n) | 2 | SelfAttn | 7 | GCN | ✓ | 44.30 ±3.54 | *The number of decoder layers* |
| (o) | 2 | SelfAttn | 8 | GCN | ✓ | 43.20 ±2.95 | *The number of decoder layers* |
| (p) | 2 | SelfAttn | 12 | GCN | ✓ | 43.38 ±2.94 | *The number of decoder layers* |
| (q) | 1 | SelfAttn | 1 | GCN | ✗ | 34.98 ±3.77 | *Single layer for both encoder and decoder* |
| (r) | 1 | SelfAttn | 1 | GAT | ✗ | 36.49 ±3.41 | *Single layer for both encoder and decoder* |

To demonstrate the effectiveness of *NATR*, we report the comparison of models with various conditions in Table 6 of the main text. The illustration to help understanding each model in Table 6 of the main text is shown in Figure 9. The $MLP$ model, (A) produces the node-level representation by mixing attribute representations **equally**, without the message propagation. The (A) model suffers from the intra-node dilution but not the inter-node dilution with high value of the inter-node dilution factor ($\delta_{\text{Agg}}^{\text{inter}}(v) = 1$ for all nodes) and the low value of the intra-node dilution factor $\delta_v^{\text{intra}}(t) = 1/|\mathcal{T}_v|$. The $NATR_{MLP}$ models, (D) and (E) also do not propagate node representations, but instead mix attribute representations according to attention coefficients (importance of each attribute). This comparison also reveals that *NATR* effectively alleviates the intra-node dilution.

We also provide comprehensive ablation studies with various configurations in Table9. The effectiveness of the node-level aggregation in *NATR* is explained by (a)→(ref1), with a performance gain about 7.49 (21.5%). It implies the importance of both preserving attribute-level representation and aggregating node-level representation. The effectiveness of the encoder (i.e. the correlation between attributes) is demonstrated by (c)→(a) (+1.63, 4.9%) and (b)→(h) (+1.12, 2.7%). The intermediate loss is crucial as the number of decoder layers increases, demonstrated by (d)→(ref1) (+19.88, 88.4%) and (e)→(ref2) (+6.77, 20.5%). We fix the number of encoder layers as two and the number of decoder layers less than six when comparing to MPNNs under controlled conditions in the paper. However, (f)-(j) and (m)-(p) show the potential of *NATR* to boost performance.

Table 10: P: the number of parameters, S: model size (actual disk size in MB), I: inference time (ms) are reported with $d = 128, d_{FFN} = 512, N_{ENC} = 2$ on Computers dataset. The average of the practical inference time is measured 10k times on GPU - RTX 8000.

| | 2 LAYERS | | | 3 LAYERS | | | 4 LAYERS | | | 5 LAYERS | | |
|---|---|---|---|---|---|---|---|---|---|---|---|---|
| | P | S | I | P | S | I | P | S | I | P | S | I |
| $NATR_{GCN}$ W/ MLP ENC | 830,592 | 3.18 | 16.78 | 1,062,144 | 4.07 | 24.98 | 1,293,696 | 4.96 | 33.19 | 1,525,248 | 5.85 | 41.41 |
| $NATR_{GCN}$ W/ SelfAttn ENC | 1,194,368 | 4.58 | 17.60 | 1,409,408 | 5.40 | 25.80 | 1,624,448 | 6.23 | 33.99 | 1,839,488 | 7.05 | 42.19 |
| $NATR_{GAT}$ W/ MLP ENC | 831,104 | 3.19 | 22.45 | 1,062,912 | 4.08 | 34.04 | 1,294,720 | 4.97 | 45.36 | 1,526,528 | 5.86 | 56.23 |
| $NATR_{GAT}$ W/ SelfAttn ENC | 1,194,880 | 4.58 | 23.27 | 1,410,176 | 5.41 | 34.48 | 1,625,472 | 6.24 | 46.04 | 1,840,768 | 7.06 | 56.80 |
| $NATR_{SGC}$ W/ MLP ENC | 813,824 | 3.12 | 18.10 | 1,028,608 | 3.94 | 26.10 | 1,243,392 | 4.76 | 34.35 | 1,458,176 | 5.59 | 42.38 |
| $NATR_{SGC}$ W/ SelfAttn ENC | 1,177,600 | 4.51 | 18.95 | 1,375,872 | 5.27 | 27.10 | 1,574,144 | 6.03 | 35.13 | 1,772,416 | 6.79 | 42.98 |

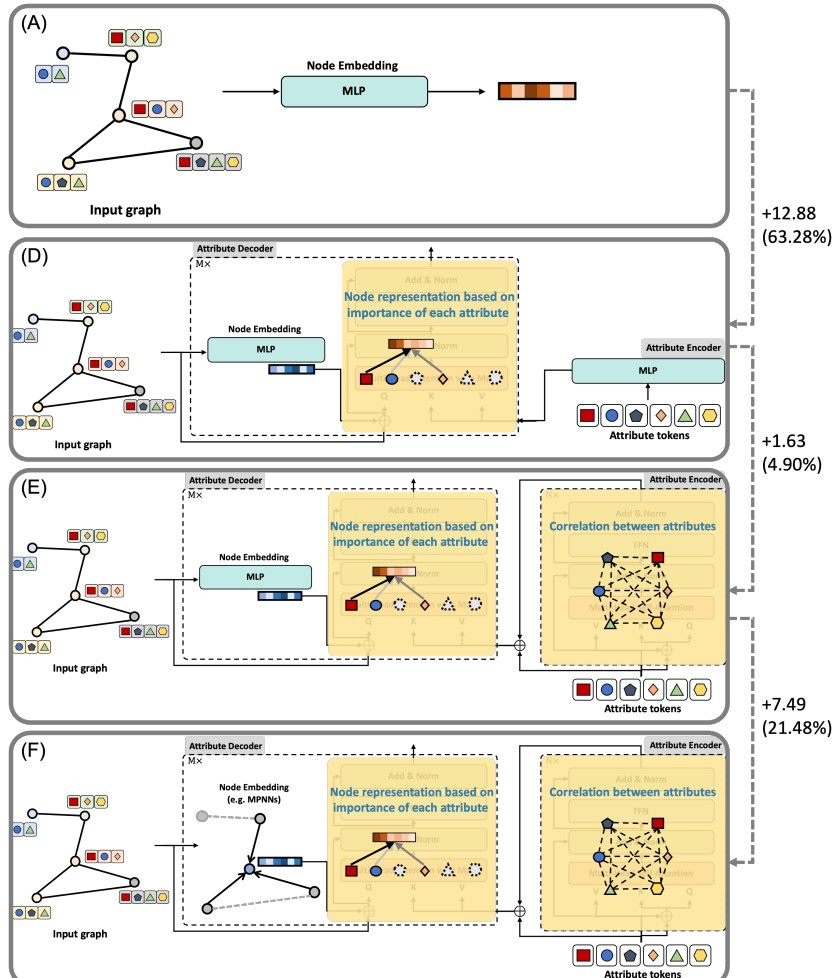

Figure 9: The illustration for models in Table 5 of the paper. Compared to *MLP* model (A), the $NATR_{MLP}$ with *MLP* encoder (D) alleviates the intra-node dilution by mixing attribute representations based on the attention coefficients between the node-level representation and attribute representations, improving performance about 12.88 (63.28%). The model (E) considers the correlation between attributes by using $\mathrm{SelfAttn}$ as the attribute encoder and improves performance about 1.63 (4.90%) over the (D) model. The complete model (F), $NATR_{GCN}$ is equipped with MPNNs (*GCN*) as a node embedding module in the attribute decoder. (F) produces node-level representations on the context of graph structure and uses them as queries for attribute representations. This brings the performance gain about 7.49 (21.48%) over the model (E).

## H  COMPLEXITY

In this section, we analyze the computational complexity which can be a limitation of *NATR* due to quadratic computation. The computational complexity with omitting $d$ and $l$ is depending on $O(|\mathcal{T}|^2)$ for the attribute encoder and $O(|\mathcal{V}||\mathcal{T}|)$ for the attribute decoder, and $O(|\mathcal{V}| + |\mathcal{E}|)$ for MPNNs. Even though the complexity of the attribute encoder is quadratic to the number of attributes, the number of attributes is relatively smaller than the number of nodes as shown in Table 7 because the attributes are specific information about the nodes, and the number of attributes is rarely increased unlike the number of nodes which can increase with the expansion of the graph. As seen in Table 10, the attribute encoder with $\mathrm{SelfAttn}$ has a slightly higher practical computation time of about 1 ms compared to the encoder with *MLP*.

# I   PLUG-IN VERSION WITH THE PRE-TRAINED MODEL

As described the main text, we introduce the plug-in version as a variant of *NATR* for single-layer models such as $SGC$. The plug-in version architecture can be also applied in scenarios where the node embedding model already exists. For example, if a trained model is already being utilized for service execution and it is infeasible to re-train or incorporate it into the decoder of *NATR*, it can be utilized as a query generator of the plug-in version. As shown in Figure 11, we train the *GCN* (green line) with the best configuration and attach it to the plug-in version of *NATR* (orange line). We can find that the $NATR_{GCN}^{pre}$ which is trained from the

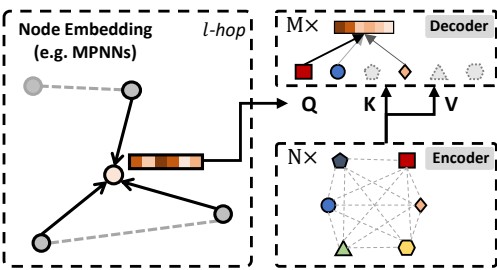

Figure 10: Plug-in version of *NATR*. The node embedding module is operated separately from the decoder to generate queries.

pre-trained *GCN* converges faster than the $NATR_{GCN}^{scratch}$ which is trained from scratch. It implies that *NATR* architecture is beneficial even for the pre-trained models. The $NATR_{GCN}^{pre}$ model and the $NATR_{GCN}^{scratch}$ show the performance on test set 38.830 ±4.026 and 38.171 ±3.629, respectively.

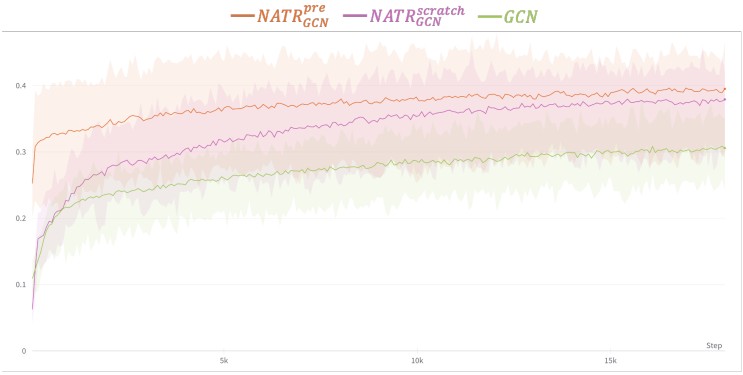

Figure 11: Hits@20 performance on validation set in the Computers dataset. The orange line (top) and the purple line (middle) indicate the performance of *NATR* trained from a pre-trained model and from scratch, respectively. The green line (bottom) indicates the performance of the *GCN* model.

# J   RELATED WORKS

## J.1   GRAPH TRANSFORMERS

*NATR* is a transformer that can be applied to graph-structured data. Recently, there have been numerous endeavors to utilize the benefits of transformer architecture in the contexts of node embedding and graph embedding such as Graphormer, GRPE, EGT, SAN, TokenGT, and GraphGPS (Ying et al., 2021; Dwivedi & Bresson, 2021; Kreuzer et al., 2021; Park et al., 2022; Hussain et al., 2022; Kim et al., 2022; Rampášek et al., 2022). However, the comparison with graph transformers is out of main scope in this work because we focus on the alleviation of over-dilution phenomenon. The transformer in *NATR* accepts attribute-level representations as inputs (specifically, key and value) to alleviates the over-dilution issue while others take node-level representations for long-range dependencies. It is noteworthy that our model, owing to its general architecture, can be seamlessly integrated with the graph transformers. As reported in Table 8, we exploit Graphormer (Ying et al., 2021) as a node embedding module in the decoder of $NATR_{Graphormer}$. The graph transformers, which are used to generate graph-level representations, can utilize *NATR* to generate node-level tokens rather than using graph transformers as a node embedding module of *NATR*. In this manner, existing graph transformers and *NATR* can be employed in a complementary fashion.

