# OpenReview forum: "Understanding and Tackling Over-Dilution in Graph Neural Networks"
_ICLR.cc/2024/Conference — Submitted to ICLR 2024_

### Official Review · Reviewer_V15d · 2023-10-18

**Soundness:** 2 fair
**Presentation:** 3 good
**Contribution:** 2 fair
**Rating:** 6
**Confidence:** 5

**Summary:**

1. The paper identifies a limitation in Message Passing Neural Networks (MPNNs) related to the handling of node attributes in graph datasets, specifically in how attributes lose significance during attribute transformation and feature aggregation stages.
2. Even single-layer MPNNs weaken node representations by combining attributes equally, leading to a phenomenon called over-dilution, where node features become diluted and less informative.
3. To address this issue, the paper introduces Node Attribute Transformer (NATR), a transformer designed to operate on node attributes, producing more informative node representations and mitigating the problem of over-dilution in MPNNs.

**Strengths:**

1. Figures 1 and 4 provide clear visual distinctions between the over-dilution phenomenon and existing phenomena of over-smoothing and over-squashing.
2. Over-dilution provides an original perspective, broadening the scope of MPNN limitations.
3. The phenomenon is assessed by dividing it into two sub-phenomena: intra- and inter-node dilution, along with the introduction of corresponding factors.
4. NATR is incorporated with existing MPNNs to demonstrably counteract the effects of over-dilution on node classification and link prediction.

**Weaknesses:**

1. The significance of the paper can be strengthened by exploring graph datasets, at least synthetic data, and ideally real-world examples, exhibiting over-dilution in deep layers without the presence of other phenomena, e.g., over-smoothing.
2. Over-dilution is detected in the very first layer of MPNNs, distinguishing it from other phenomena, but there are no convincing real-world experiments to support the notable implications of this single-layer over-dilution.
3. Over-correlation, documented in existing studies [Liu et al., 2023, Jin et al., 2022], aligns with over-dilution in the realm of *preservation of attribute-level information*, necessitating a comprehensive discussion to discern their nuanced differences.

References
* [Liu et al., 2023]: Enhancing Graph Representations Learning with Decorrelated Propagation, KDD'23
* [Jin et al., 2022]: Feature Overcorrelation in Deep Graph Neural Networks: A New Perspective, KDD'22

**Questions:**

1. What were the criteria for selecting the five real-world graph datasets, shown in Table 7, for studying over-dilution?
2. Related to the previous question, is it the case that MPNNs were more susceptible to over-dilution on these datasets than other existing datasets?
3. What characteristics were considered when choosing these datasets to ensure they accurately represent over-dilution without the interference of other phenomena, e.g., over-smoothing, over-correlation?
4. How was it ensured that the superior performance achieved by NATR in settings with 4 or 5 layers, as demonstrated in Table 3, is solely attributed to reduced over-dilution and not a result of significantly mitigating *possibly more severe phenomena* such as over-smoothing, over-correlation?
5. Were there insights into potential real-world scenarios or applications where single-layer over-dilution could have significant consequences?

---

> ### Author Response · Authors · 2023-11-15
>
> With sincerely thank you for recognizing our contributions, we've carefully addressed each point in the review to further elevate our work.
>
> ___
>
>
> **W1: The significance of the paper exhibiting over-dilution in deep layers without the presence of other phenomena, e.g., over-smoothing**
>
> Identifying over-dilution in deep layers without interference from other phenomena like over-smoothing is challenging due to the shared underlying cause, such as the size of the receptive field.
>
> Therefore, our focus was on demonstrating the impact of over-dilution on performance in a single layer, where other phenomena are less prevalent.
>
> For a more detailed explanation, please refer to our response to W2 & Q4.
>
> ___
>
>
> **W2 & Q4: Experimental evidence of NATR’s effectiveness in addressing the over-dilution issue**
>
>
> - intra-node dilution
>
> We may not have emphasized this sufficiently in the text, but Table 5 clearly demonstrates the effectiveness through a comparison between (A)- $\textit{MLP}$ and (D)- $\textit{NATR}_{\text{MLP}}$.
>
> In both models, message passing is not utilized, ruling out the issue of over-smoothing.
>
> In this context, the performance improvement observed with NATR can be attributed to its ability to resolve over-dilution at the intra-node level, regardless of the over-smoothing issue.
>
> It's important to note that even with an increased number of parameters in the MLP model, there was no observed improvement, further emphasizing NATR's unique capability in addressing over-dilution.
>
> - inter-node dilution
>
> We demonstrated single-layer over-dilution using the Computers dataset as an example. Here, a typical node has 204 attributes (median value) and 19 neighbors (median degree). In this case, each attribute's representation dilutes to approximately 0.025% per layer with either the *mean* or *sum* aggregation operator.
>
> Table 9 in the Appendix reveals that NATR significantly outperforms MPNNs even with a single layer, as evident in examples (q) and (r).
>
> Typically, issues like over-smoothing, over-squashing, and over-correlation occur after multiple layers of message passing.
>
> Hence, this demonstrates NATR's effectiveness in mitigating mainly over-dilution, both at the intra-node and inter-node levels.
>
>
> ___
>
>
> **W3: Difference with over-correlation**
>
>
> According to Jin et al., the over-correlation refers to the phenomenon where the feature representations learned by the **stacked** network become highly correlated across different **dimensions**.
>
> In the node-level representation of MPNNs, each dimension does NOT mean each attribute representation (for initial node representation, it is a summed value of the corresponding dimension from attribute-level representations).
>
> Therefore, the concept of over-dilution is distinct from overcorrelation.
>
> Also, the intra-node dilution can occur even at the single-layer while overcorrelation is observed when stacking layers.
>
> However, we agree that it would be good to discuss its connection to overcorrelation because both of them are unique perspectives to construct more informative node representations.
>
>
> ___
>
>
> **Q1, Q2, Q3: the criteria for selecting the datasets**
>
>
> We selected benchmark datasets that contain attribute information.
>
> OGB offers dense node features (or one-hot vector) rather than explicit attribute indicators.
>
> For the ogbl-ddi dataset, we were able to extract attributes in a fingerprint format (binary vector) using RDKit, which is why it has been included in our benchmark dataset selection.
>
> If there is a larger dataset that contains attribute information, we will include it as our benchmark dataset.
>
> ___

---

> > ### Author Response · Authors · 2023-11-15
> >
> > **Q5: Insights into potential real-world scenarios about over-dilution**
> >
> > Indeed, there were insights into real-world scenarios where the over-dilution could have significant consequences, particularly in drug-drug interaction networks (e.g. OGB-DDI).
> >
> > In these networks, drugs are represented as nodes, and their chemical substructures, which are crucial for predicting interactions between drugs, are used as attributes.
> >
> > (each attribute corresponds to a specific substructure of the drugs)
> >
> > However, unlike the approach in MPNNs, it is important to note that not all attributes are equally significant.
> >
> > For example, Aspirin and Ibuprofen, despite sharing a common substructure (attribute), exhibit distinct properties, especially in terms of efficacy and side effects.
> >
> > In such cases, this attribute does not differentiate the two drugs, making exclusive attributes crucial.
> >
> > Therefore, in constructing node-level representations, it is vital to carefully consider each attribute in the intra-node context.
> >
> > This need to focus on individual attributes for informative node representations is particularly important in scenarios like DDI networks, where overlooking unique attributes due to over-dilution can lead to inaccurate predictions and analyses.
> >
> > ___
> >
> > We appreciate your thorough review and hope our detailed answers have resolved your queries.
> > If there is anything else that needs further explanation, we are more than willing to provide additional information.

---

> > > ### Comment · Reviewer_V15d · 2023-11-23
> > > **Thanks for the responses**
> > >
> > > Thanks for the rebuttal. After carefully examining all the reviews and their responses, my level of confidence regarding the evaluation has risen.

---

### Official Review · Reviewer_LoKp · 2023-10-29

**Soundness:** 3 good
**Presentation:** 2 fair
**Contribution:** 2 fair
**Rating:** 5
**Confidence:** 3

**Summary:**

This paper introduces a new limitation of Message Passing Neural Networks (MPNNs) (i.e., over-dilution). It shows two types of dilutions: intra-node dilution and inter-node dilution considering 1) the equal weight combination for attributes within each node, 2) the information from neighbors is diluted through aggregation. The authors also provide formal definitions of these concepts. To mitigate the problem, they propose a transformer-based method (NATR) by considering adaptively merging attribute representations. The experiments are conducted for link prediction and node classification tasks, showing the better performance of NATR.

**Strengths:**

1. The motivation to adaptively utilize attributes for each node is sound.
2. The analysis about dilution factors and the formal definitions have some merits.
3. The improvements in some datasets are impressive.

**Weaknesses:**

1. While the authors conduct the experiments on both link prediction and node classification, they only use three datasets (i.e., computers, photo, and cora ML) for node classification. OGB datasets for node classification are not included. I would like to see some results on ogbn-arxiv or ogbn-product. Even if the model may not perform well on these datasets, I suggest the author provide some analyses or insights about what kind of datasets would benefit more by using the proposed model.
2. To me, it's not very clear for some parts of the analysis (e.g., Sec 6.2). In Sec 6.2, the author investigates the performance for nodes with bottom 25% and top 25% of inter-dilution scores. But for the base model and the version with NATR, the formula of $\delta^{inter}_{Agg}(v) $ should be different? For NATR, it uses Eq (11). For GCN, it uses Eq (7). So, I wonder if the nodes are separated by only considering the original inter-dilution factor (i.e., Eq (7))?
3. The proposed method is claimed to solve both intra-dilution and inter-dilution. For intra-node dilution, the method can assign larger weights to more important attributes. However, it is unclear to me how the method addresses the inter-node dilution. I would suggest the author elaborate more on this part.

**Questions:**

1. Based on my understanding, in the proposed method, each attribute has its own learnable representations. In this case, the number of learnable parameters will increase compared with previous MPNN baselines (e.g., GCN). I wonder whether the performance improvement is mainly from the increased number of learnable parameters. I would like to see some analyses in this regard.
2. I am curious how the performance would change with different numbers of attributes in a graph. Is there any trend for this?
3. Considering the motivation of avoiding the combination of attributes with different weights, I wonder if feature selection methods can help to alleviate this problem.

---

> ### Author Response · Authors · 2023-11-15
>
> We appreciate your feedback and would like to clarify any contributions that we may not have sufficiently emphasized in the current version.
>
> ---
>
>
>
> **W1: Experiments on other datasets such as ogbn-arxiv or ogbn-product**
>
> Since Reviewer HxeZ also pointed this out, it seems we missed adding more explanation about this point.
>
> We chose benchmark datasets containing attribute information, as our primary focus is on attribute-level representation.
>
> However, OGB offers dense node features rather than explicit attribute indicators.
>
> - ogbn-arxiv : 128-dimensional feature vector obtained by averaging the embeddings of words
> - ogbn-product : 100-dimensional feature vector obtained by PCA of bag-of-words features
>
> In the case of the ogbl-ddi dataset, we successfully extracted attributes in a fingerprint format (binary vector) using RDKit and DrugBank DB.
>
> If larger datasets become available that include explicit attribute information, we are certainly prepared to include them in our benchmark datasets.
>
> ---
>
>
>
> **W2: The implication of the analysis in Section 6.2**
>
> You are correct. Indeed, the subsets of the bottom 25% and the top 25% were separated following Eq (7) (as two-hops version) and we used the fixed subsets for a fair comparison between MPNNs and NATR.
>
> As presented in Table 4, we delve deeper into how the NATR model exhibits performance improvements when combined with MPNNs.
>
> GCN, GAT, and SGC demonstrate notably lower performance on over-diluted nodes (bottom 25%).
>
> For the same subsets, NATR shows a more significant improvement in performance on over-diluted nodes than less-diluted nodes.
>
> This distinction highlights the effectiveness of NATR in mitigating over-dilution issue and illustrates the benefits of our proposed approach in enhancing model performance where MPNNs may struggle.
>
>
> ---
>
>
> **W3: How the method addresses the inter-node dilution**
>
> To address inter-node dilution, NATR's attribute decoder integrates attribute representations across all layers.
>
> This integration helps prevent the dilution of a node's own information in the context of its neighboring nodes.
>
> The final node-level representation of node $v$, $\tilde{H}_v^{(M)}$ is calculated by combining two representations: $H_v^{(M)}$ for graph context information and $O_v^{(M)}$ for node $v$’s specific information, exclusively.
>
> This approach effectively mitigates inter-node dilution, maintaining distinct node information within the graph context, as described in Eq (11) and Figure 2 (b).
>
> ---
>
>
>
> **Q1: The number of parameters of each model and the effect of additional parameters**
>
>
> It is an acknowledged fact that the use of a transformer architecture in NATR leads to additional parameters and increased computational complexity.
>
> However, it is important to note that simply increasing the number of parameters in existing MPNNs does not yield performance improvements comparable to those achieved by NATR.
>
> You can see the degradation of performance in MPNNs when the number of parameters (layers) increases in Table 3.
>
> Significantly, NATR leverages the transformer architecture to effectively address the issue of over-dilution.
>
> This not only enhances performance but also facilitates the use of deeper layers (please see Table 9 as well), demonstrating the utility and efficacy of our approach.
>
>
> ---
>
>
> **Q2: the performance would change with different numbers of attributes in a graph**
>
>
> The performance can be affected not only by the number of attributes but also by the complicated property of each attribute.
>
> Therefore, it is hard to figure out the trend of the correlation between performance and the number of attributes.
>
> ---
>
>
>
> **Q3: feature selection methods can help to alleviate this problem**
>
> We agree that feature selection methods can be beneficial in mitigating the issue of combining attributes with varying levels of importance. However, these methods typically apply uniform weighting across all nodes.
>
> Nevertheless, it's crucial to recognize that the significance of attributes can vary from node to node.
>
> For instance, while attribute 't' may be important for node A, it might not hold the same level of significance for node B.
>
> Our NATR model addresses this by employing a graph context-aware attribute decoder.
>
> This decoder utilizes node-level representations as queries, which enables the assignment of greater weight to the more significant attribute representations for each individual node.
>
> This approach ensures that the importance of attributes is determined in the context of each node's specific role and relationship within the graph.
>
>
> ---
>
> Thank you again for your valuable comments.
> We have attempted to address each of your questions thoroughly, enhancing the clarity of our paper's contributions.
> Please let us know if there are points that require further explanation!

---

### Official Review · Reviewer_HxeZ · 2023-11-04

**Soundness:** 2 fair
**Presentation:** 2 fair
**Contribution:** 2 fair
**Rating:** 5
**Confidence:** 3

**Summary:**

The paper discusses a recent challenge in the field of Graph Neural Networks (GNNs), particularly focusing on Message Passing Neural Networks (MPNNs), and introduces the issue of "over-dilution" where node attribute information is diminished in the final representation due to excessive aggregation from many attributes (intra-node dilution) or overwhelming information from neighboring nodes (inter-node dilution). The authors propose a novel transformer-based architecture that treats attribute representations as tokens, which, unlike being a replacement, is an augmentation to existing MPNNs. This model aims to preserve attribute-level information more effectively by using attention scores to weigh attribute representations in the context of the aggregated node-level representation. The paper claims to contribute a new perspective on the problem of over-dilution by defining and analyzing it, which is distinct from the commonly discussed limitations of MPNNs such as over-smoothing, over-squashing, and over-correlation. The proposed transformer-based solution is theoretically and empirically validated for its efficiency in maintaining attribute-level information within graph-structured data representations.

**Strengths:**

* The paper introduces the concept of over-dilution, a novel perspective in the study of GNNs, particularly MPNNs, that goes beyond the well-studied limitations of over-smoothing, over-squashing.
* The proposed transformer-based architecture is not only theoretically grounded but also empirically tested, providing a strong case for its effectiveness in combating the over-dilution problem. This dual approach enhances the credibility of the findings.

**Weaknesses:**

* Experiments are not complete.
* The story of this paper is weird. I don't know why the author include over-smoothing and over-squashing as a story and don't do any comparison between over-dilution and them.

**Questions:**

* For baselines, I think GCNII can be moved into the main paper and can you do it on all datasets? Because GCNII can alleviate over-smoothing, which I think maybe relevant to the paper.
* Also, How is GCNII experiments done? Have you tried hyperparameter searching on it?
* For datasets, even the authors state that the complexity is acceptable, the datasets the paper used are all small datasets. Can you provide results and time comparison with backbone on some larger datasets? like ogb-arxiv or ogb-citation2(maybe too large, ogb-ppa can also be a good choice).
* Can the authors provide number of parameters of each model with backbone? How to know the improvement is not the result of adding new parameters in the transformer?
* Can the authors provide details on what's the relationship between over-smoothing,over-squashing and over-dilution? Theoretically and empirically?

---

> ### Author Response · Authors · 2023-11-15
>
> Thank you for the very detailed feedback, it has been instrumental in refining our work and highlighting its strengths.
>
>
> ---
>
>
> **W2 & Q5: Clarification on Over-Smoothing, Over-Squashing, and Over-Dilution**
>
> We realize we may not have delved into the details as thoroughly as needed, although we have compared these concepts in Figure 1, we'll expand the discussion in the main text.
>
> Appendix Section A and Figure 4 provide further details both conceptually and theoretically.
>
> Over-dilution presents a novel and comprehensive concept, while being related to over-smoothing and over-squashing in terms of information distortion in graph structures.
>
> - Unique perspective on intra-node dilution
>
> We acknowledge that our initial presentation may have overlooked the unique aspects of intro-node dilution.
> Unlike over-smoothing and over-squashing, which are primarily concerned with node-level representation, over-dilution encompasses both node-level and attribute-level representations, crucial for forming informative representations on graphs.
> This phenomenon can arise at the intra-node level due to an excess of attributes within nodes, independent of the triggers for over-smoothing and over-squashing.
>
> - Comparison with over-smoothing
>
> In contrast to over-smoothing, where node representations become increasingly similar due to the exchange of information across multiple hops, over-dilution can arise even in single-layer aggregations, as demonstrated in our primary discussion.
> Take, for example, a star graph structure, which is essentially a tree with a central node and several leaf nodes, each linked to the central node by a single edge.
> Following a single message-passing step, the central node may experience over-dilution, but over-smoothing among nodes is less likely to occur since the leaf nodes do not share features with one another.
>
>
> - Comparison with over-squashing
>
> Regarding over-squashing, we now understand that our initial explanation may have been too concise. Unlike over-squashing, which deals with long-range information propagation between nodes, over-dilution focuses on the attenuation of information at individual nodes, observable even in the first layer based on aggregation coefficients.
> This distinction is evident in our analysis, where over-dilution is shown to be a prevalent issue even in short-range interactions, challenging the traditional understanding of information propagation in graph neural networks.
> Conceptually, over-dilution focus on the preservation of information (formally $\partial h_x^{(l)}/\partial h_{\color{Red}x}^{(0)}$), while over-squashing focus on the propagation ( $\partial h_x^{(l)}/\partial h_{\color{Red}y}^{(0)}$).
>
>
> ---
>
>
> **Q1 & Q2: Comparison with GCNII**
>
> First and foremost, it's essential to clarify that NATR works as a complementary model to any node embedding module, not as a competitor to a specific standalone model.
>
> Although GCNII is effective in mitigating over-smoothing, this does not necessarily make it superior to NATR, as NATR is specifically tailored to address the issue of over-dilution.
>
> When equipped with GCNII as its node embedding module, $\textit{NATR}_{\text{GCNII}}$ could potentially yield a more informative node-level query in the attribute decoder.
>
> In Table 8 and Section F of the Appendix, we present the performance of NATR using different node embedding modules, demonstrating its compatibility rather than positioning it as a competitor.
>
> This experiment involved a hyperparameter search, the details of which are outlined in Section E of the Appendix.
>
> ---
>
>
>
>
> **Q3: Experiments on larger datasets such as ogb-arxiv, ogb-citation2, or ogb-ppa**
>
> Your point regarding dataset selection is well-taken.
>
> Our choice to use benchmark datasets with explicit attribute information was intentional, as we aimed to highlight NATR's capabilities in handling such data.
>
> For OGB, we encountered datasets offering dense node features or one-hot vectors rather than explicit attribute indicators:
>
> - ogbn-arxiv : 128-dimensional feature vector of word embeddings
> - ogbl-ppa : one-hot vector
> - ogbl-citation2 : 128-dimensional word2vec features
>
> In the case of the ogbl-ddi dataset, we successfully extracted attributes in a fingerprint format (binary vector) using RDKit and DrugBank DB.
>
> If larger datasets that include explicit attribute information become available, or if you are aware of any, we would definitely be happy to consider them!
>
>
> ---

---

> > ### Author Response · Authors · 2023-11-15
> >
> > **Q3 & Q4: The time comparison and the number of parameters of each model and the effect of additional parameters**
> >
> > In Table 10 of Appendix, we selected the largest dataset among benchmark datasets we used, the Computers dataset, to comprehensively report the number of parameters, model size (actual disk size), and inference time for each model, broken down by the number of layers.
> >
> > It is a fair point that the use of a transformer architecture in NATR leads to additional parameters and increased computational complexity.
> >
> > However, it is important to note that simply increasing the number of parameters in existing MPNNs could not yield performance improvements comparable to those achieved by NATR.
> > In fact, this comparison shows well that NATR is effective.
> >
> > ---
> >
> > Thank you again for the constructive feedback!
> > We have endeavored to address all of your points and hope our paper now better reflects its value and contribution.
> > If anything remained unclear, we are eager to address them to improve our manuscript further!

---

> > > ### Comment · Reviewer_HxeZ · 2023-11-21
> > > **Response to the Authors**
> > >
> > > I appreciate the authors' response.  I'll raise my score to 5.

---

> > > > ### Author Response · Authors · 2023-11-21
> > > >
> > > > Dear Reviewer HxeZ,
> > > >
> > > > Thank you so much for your timely reply and for recognizing our rebuttal!
> > > >
> > > > We greatly appreciate your decision to raise the score.
> > > >
> > > > Of course, we respectfully acknowledge your evaluation.
> > > >
> > > > Please let us know if you have any specific suggestions for final improvements.
> > > >
> > > > Best regards,
> > > >
> > > > Authors

---

### Official Review · Reviewer_gjvj · 2023-11-10

**Soundness:** 2 fair
**Presentation:** 3 good
**Contribution:** 2 fair
**Rating:** 6
**Confidence:** 4

**Summary:**

This paper proposes to study a new pheonomenon named over-dilution in message passing neural networks (MPNNs). It refers to the diminishing importance of a node's information in the final node representations learned by the neural networks. The authors propose NATR to address the proposed over-dilution problem. The key idea is to learn an attribute encoder and then train another transformer-based attribute decoder where Q is the node embeddings from MPNNs and K, V are the embeddings output by the attribute encoder.

**Strengths:**

S1. Interesting new perspective to study the limitation of MPNNs.

S2. Improved performance in tasks like link prediction and node classification.

**Weaknesses:**

Please see questions below.

**Questions:**

Q1. I am confused by Eq. (3). Isn't $z_t$ the same as the $t$-th value in $h_v^{(0)}$? Why is it $h_v^{(0)}$ rather than $h_v^{(k)}$ in some hidden layer $k$?

Q2. Still about Eq. (3): this essentially measure some normalized correlation between one attribute and another attribute, i.e., how a infinitesimal perturbation on attribute $t$ would affect other attributes in node features $h_v^{(0)}$. It would be good to discuss the its connection to overcorrelation by Jin et al.

Q3. Definition 3.2 is the same as Xu et al., so it is necessary to cite it in Definition 3.2.

Q4. Hypothesis 2 seems related to degree fairness learned in several papers [1, 2, 3]. When the node degree is high, after normalization, the aggregation weight $\alpha_{v, v}$ will be smaller than the sum of all other edge weights. It would be good to discussion some intrinsic connection to this line of work.

Q5. Hypothesis 3 seems to be very related to over-squashing by Topping et al. It would be good to have more in-depth discussion on the difference between Hypothesis 3 and over-squashing.

Q6. To me, it feels that NATR would help when the number of layers increases. But it seems the MPNNs used in experiments are pretty shallow. What would happen if we increase the layers to a larger number? How would NATR perform if we equip it with deep graph neural networks like RevGCN [4]?

Q7. The over-dilution seems like some combination of feature correlation (Definition 3.1) and over-squashing (Definition 3.2, Hypothesis 3) to me. It would be better to discuss the difference between the over-dilution and these two scenarios.


**References**

[1] Tang, Xianfeng, et al. "Investigating and mitigating degree-related biases in graph convoltuional networks." Proceedings of the 29th ACM International Conference on Information & Knowledge Management. 2020.

[2] Kang, Jian, et al. "Rawlsgcn: Towards rawlsian difference principle on graph convolutional network." Proceedings of the ACM Web Conference 2022. 2022.

[3] Liu, Zemin, Trung-Kien Nguyen, and Yuan Fang. "On Generalized Degree Fairness in Graph Neural Networks." arXiv preprint arXiv:2302.03881 (2023).

[4] Li, Guohao, et al. "Training graph neural networks with 1000 layers." International conference on machine learning. PMLR, 2021.

---

> ### Author Response · Authors · 2023-11-15
>
> We want to express our gratitude for the thorough review and have addressed each concern in detail.
>
> ---
>
>
> **Q1: Clarification on notations.**
>
>
> In MPNNs, the initial node-level representation $h_v^{(0)} \in \mathbb{R}^d$ is a summation (or averaging) of attribute representations $z_t \in \mathbb{R}^d$, such that  $h_v^{(0)} = \sum_{t \in \mathcal{T}_v} z_t$.
>
> Therefore, the $t$-th value of $h_v^{(0)}$ is NOT the same as $z_t$; instead, each dimension of $h_v^{(0)}$ is a summed value of the corresponding dimension from attribute-level representations.
>
> We define the dilution phenomenon as two **cascaded** sub-phenomena as intra-node dilution→ inter-node dilution.
>
> The intra-node dilution occurs when constructing the initial node-level representation $h^{(0)}_v$ while the inter-node dilution occurs when aggregating node-level representations (for $h^{(k)}_v$).
>
> Therefore, we use the initial node-level representation $h^{(0)}_v$ (NOT $h^{(k)}_v$) in Eq. (3) to describe the intra-node dilution.
>
>
> ---
>
>
> **Q2 & Q7: Connection to overcorrelation (Jin et al.).**
>
> According to Jin et al., the over-correlation refers to the phenomenon where the feature representations learned by the **stacked** network become highly correlated across different **dimensions**.
>
> As we answered in Q1, each dimension of node-level representation does NOT mean each attribute representation (it is a summed value of the corresponding dimension from attribute-level representations.).
>
> Therefore, the concept of over-dilution (specifically, the intra-node dilution) is distinct from overcorrelation.
>
> Also, the intra-node dilution can occur even at the single-layer while overcorrelation is observed when stacking layers.
>
> However, we agree that it would be good to discuss its connection to overcorrelation because both of them are unique perspectives to construct more informative node representations.
>
> We will incorporate this discussion into the revised version of our paper.
>
>
> ---
>
>
> **Q3, Q5, Q7: Difference with the influence distribution (Xu et al.) and over-squashing**
>
>
> While the influence distribution with Jacobian (Xu et al.) serves as a clear framework to analyze limitations of MPNNs, as utilized in the over-squashing paper (Topping et al.), our Definition 3.2 is NOT the same.
>
> The numerator in our Definition 3.2 differs to reflect our unique perspective on how information is preserved within nodes (i.e. $\partial h_x^{(l)}/\partial h_{\color{Red}x}^{(0)}$), in contrast to the approach of Xu et al. and Topping et al., which focus on information propagation between nodes (i.e. $\partial h_x^{(l)}/\partial h_{\color{Red}y}^{(0)}$) .
>
> (we have appropriately mentioned and cited the use of the influence distribution with Jacobian in our work.)
>
>
> For a detailed comparison with over-squashing, please refer to Figure 1 in the main text and Section A with Figure 4 in Appendix.
>
>
> ---
>
>
> **Q4: Relationship between degree fairness (normalization) papers.**
>
> Thanks for the suggestion of the discussion about potential link between our Hypothesis 2 and the concept of degree fairness!
>
> Our Hypothesis 2 indeed provides a comprehensive framework to examine the interplay between over-dilution and aggregation weight $\alpha$ that can be defined not only by degree but also by factors like the attention coefficient, as seen in GAT models.
>
> Recognizing this, we agree that it is valuable to delve into a more detailed discussion of the connections with these specific works, thereby enriching our analysis of the over-dilution phenomenon and its broader implications.
>
>
> ---
>
>
> **Q6: The performance of NATR when the number of layers increases.**
>
> As you correctly noted, NATR indeed retains its advantages even as the number of layers increases.
>
> It's important to emphasize that our primary goal is to demonstrate the effectiveness of NATR in solving the over-dilution phenomenon, rather than specifically focusing on building deep GNNs.
>
> In our main experiments, we found that just 5 layers were sufficient to saturate the size of receptive field (please refer to Figure 2 (b)) and to significantly degrade the performance of MPNN models, effectively demonstrating the efficacy of NATR in such a context.
>
> However, to further illustrate NATR's robustness with an increased number of layers, we’ve extended our experiments up to 12 layers, as shown in Table 9 of the Appendix.
>
> These additional experiments have confirmed that NATR not only maintains its performance but also improves upon it compared to the results observed with 5 layers in the main text.
>
> ---
>
> Your feedback has been invaluable in enhancing our paper.
> We believe we have addressed each point and hope our revisions have made our contributions more evident.
> Please let us know if any question still needs clarification.

---

> > ### Comment · Reviewer_gjvj · 2023-11-23
> > **Thank you for the response**
> >
> > Thank you for your response. My concerns are addressed, and I am raising my score. I believe the paper quality could be further improved if the above discussions can be incorporated into the revised version.

---

> > > ### Author Response · Authors · 2023-11-23
> > >
> > > Dear Reviewer gjvj,
> > >
> > > We are grateful for your acknowledgement of our efforts to address the concerns you raised in your review.
> > > We are pleased to know that our rebuttal has successfully met your expectations.
> > > If you have any additional questions or concerns, please feel free to share them with us, as we are dedicated to continuously enhancing the quality of our work.
> > >
> > > Authors

---

### Author Response · Authors · 2023-11-21

Dear reviewers,

We submitted our rebuttal timely so that we'd have the chance to reply to potential feedback.

We know that everyone is busy, but please let us know in case anything remained unclear.


Thank you so much for taking that time!


Authors

---

### Meta-Review · Area_Chair_ahAB · 2023-12-19

**Metareview:**

This paper identifies a new limitation of MPNNs, over-dilution, where node features/attributes become “diluted” during forward propagation. They distinguish two sources of over-dilution, intra-node dilution and inter-node dilution. Formally, they define dilution as the influence of the node itself on final representation. They also introduce a new transformer-based model that alleviates over-dilution with aggregated information over multiple layers.

Reviewers generally appreciate the proposed new perspective for the limitations of MPNNs, but most of them are still confused about the actual distinctions between over-dilution and other similar phenomena of GNNs, such as, over-smoothing and over-correlation. These well-studied previous phenomena also characterize the loss of feature information during propagation and it lacks a clear and rigorous discussion/clarification on their connections and differences. Only hypotheses of the two dilutions are proposed with no rigorous characterizations. The proposed solution of combining features from previous layers also has limited technical novelty and has been utilized by many previous works (though may for different purposes). Therefore, it remains not fully clear what is the exact difference between over-dilution and other phenomena, and what is new insight that it could bring into practice.

Based on the above considerations of lacking novelty, clarity and theoretical analyses, I would recommend rejection in its current form. The authors are encouraged to conduct more thorough theoretical analyses of the new phenomena and discuss its inherent relationships with others.

**Justification For Why Not Higher Score:**

This paper lacks a clear and rigorous discussion of the new phenomenon (especially the connection with other similar ones), which leaves unknown it is worth of being recognized by the community. More novel theoretical or empirical insights would certainly strengthen this work.

**Justification For Why Not Lower Score:**

N/A

---

### Decision · Program_Chairs · 2024-01-16

Reject